# SinkTrack: Attention Sink based Context Anchoring for Large Language Models

**Xu Liu, Guikun Chen, Wenguan Wang**[†]
The State Key Lab of Brain-Machine Intelligence, Zhejiang University

## Abstract

Large language models (LLMs) suffer from hallucination and context forgetting. Prior studies suggest that *attention drift* is a primary cause of these problems, where LLMs' focus shifts towards newly generated tokens and away from the initial input context. To counteract this, we make use of a related, intrinsic characteristic of LLMs: *attention sink* – the tendency to consistently allocate high attention to the very first token (*i.e.*, ⟨BOS⟩) of a sequence. Concretely, we propose an advanced context anchoring method, SinkTrack, which treats ⟨BOS⟩ as an information anchor and injects key contextual features (such as those derived from the input image or instruction) into its representation. As such, LLM remains anchored to the initial input context throughout the entire generation process. SinkTrack is *training-free*, *plug-and-play*, and introduces *negligible inference overhead*. Experiments demonstrate that SinkTrack mitigates hallucination and context forgetting across both textual (*e.g.*, **+21.6**% on SQuAD2.0 with *Llama3.1-8B-Instruct*) and multi-modal (*e.g.*, **+22.8**% on M3CoT with *Qwen2.5-VL-7B-Instruct*) tasks. Its consistent gains across different architectures and scales underscore the robustness and generalizability. We also analyze its underlying working mechanism from the perspective of information delivery. Our source code is available at GitHub.

## 1 Introduction

**Background.** Large language models (LLMs) have demonstrated impressive capabilities in textual and multi-modal understanding (Bevilacqua et al., 2025; Xu et al., 2024; Chen et al., 2024a; Yang et al., 2024; Li et al., 2023b; 2024; 2026). However, their deployment in critical applications is constrained by two hard, long-standing problems: *hallucination* (Chang et al., 2024; Zhang et al., 2025b; Rawte et al., 2023) and *context forgetting* (Liu et al., 2024a; Ling et al., 2023; Li et al., 2025). Hallucination is the generation of content factually inconsistent with the provided input. For instance, when asked to describe an image of buses on the road, LLM might identify "planes" that do not exist in the image (*cf.* Fig. 1a). Context forgetting occurs during long-form tasks, where LLM may lose track of initial

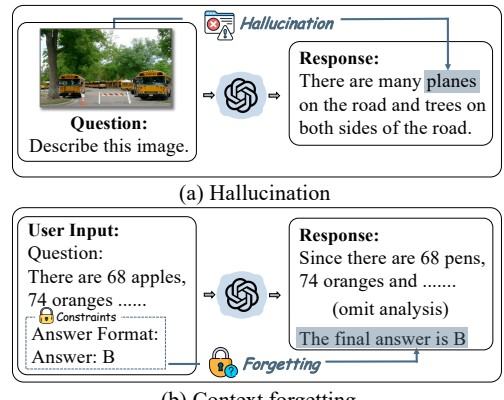

(a) Hallucination

(b) Context forgetting

Figure 1: Hallucination and context forgetting are pervasive obstacles in LLMs.

instructions, resulting in unexpected answers. In a multi-step reasoning dialogue, LLM might lose track of the initial instruction on answer formation (*cf.* Fig. 1b).

**Motivation.** Prior studies (Peysakhovich & Lerer, 2023; An et al., 2025; Liu et al., 2024a) identify *attention drift* as a direct cause of hallucination and context forgetting (*cf.* Fig. 2). As LLM generates a sequence, its attention progressively shifts towards newly created tokens, while attention on the initial tokens – which often contain the most critical input information, such as image

---

† Corresponding Author: Wenguan Wang.

features or user instructions – dramatically decays. Interestingly, we observe that this drift coexists with a countervailing phenomenon: the *attention sink* (Xiao et al., 2024; Kaul et al., 2025). While attention on most initial tokens fades, the very first token (*i.e.*, ⟨BOS⟩) intrinsically maintains a high degree of attention throughout generation, even though it is semantically sparse (*cf.* $1_{st}$ column in Fig. 2). These two patterns, one of decay and one of stability, present a synergistic opportunity. The stability of the attention sink provides a natural mechanism to counteract the information loss caused by attention drift. Based on this insight, our core idea is to transform the passive attention sink into an active mechanism for context anchoring (*i.e.*, using ⟨BOS⟩ as an information anchor), thereby mitigating the information decay caused by attention drift. Through injecting key contextual features (such as those derived from the input image or instruction) into the representation of ⟨BOS⟩, LLM remains anchored to the initial input context throughout generation.

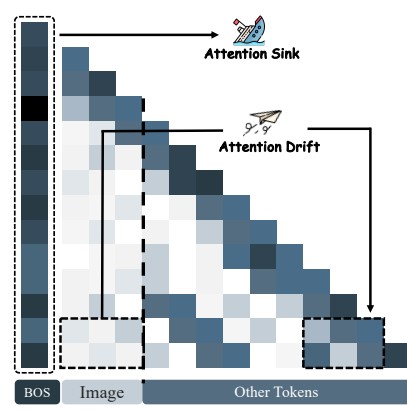

Figure 2: Attention drift and sink (the darker the color, the greater the weight).

**Methodology.** To implement the above idea, *i.e.*, using ⟨BOS⟩ as an information anchor, we make an iterative exploration. The exploration begins with a `hard injection` (§2.2), which directly replaces the Value vector of ⟨BOS⟩ in the KV cache with a mean-pooled information vector (*e.g.*, from image or instruction features). This implementation leads to a collapse, revealing that LLM's original computational flow should be preserved. We then investigate a `soft injection` (§2.3), which blends the pooled vector into LLMs using a weighted sum to respect the original flow. Despite effective, there are concerns regarding its robustness: it relies on manually tuned hyperparameters and injects all pooled representations into ⟨BOS⟩ *equally* – which might add noise and hurt performance on long contexts. Motivated by these limitations, our final solution is SINKTRACK (§2.4), which is built upon a dual-track cross-attention mechanism. In this design, one track enables ⟨BOS⟩ to adaptively query and retrieve relevant information from the context (*i.e.*, cross-attention), while a parallel self-attention track preserves LLM's native computational flow, leading to consistent performance gains.

**Merits.** Through our study of anchor formulation, injection strategies, and source integrity, we design SINKTRACK as a lightweight, adaptive enhancement with three benefits: First, it is *plug-and-play*. As a training-free, inference-time enhancement, SINKTRACK can be applied to any pre-trained LLM without the risks of fine-tuning, *e.g.*, catastrophic forgetting and model collapse. Second, it is *efficient*. Its core intervention is a one-off, lightweight operation performed prior to generation. Since the intervention can be stored in KV cache, SINKTRACK adds negligible computational overhead to the subsequent auto-regressive steps, thereby preserving LLMs' original inference speed. Third, it is *generalizable*. SINKTRACK makes use of *attention sink*, a common, intrinsic characteristic of decoder-only LLMs and multi-modal LLMs (MLLMs), to mitigate hallucination and context forgetting. This principle design makes it inherently generalizable across architectures, scales, and modalities, as validated in experiments (§3) on different LLMs and MLLMs from 3B to 12B sizes.

**Results and Analysis.** SINKTRACK demonstrates consistent performance improvements over both direct prompting and chain-of-thought (`CoT`) reasoning across various multi-modal and text-based benchmarks (§3.2). For instance, on the multi-modal M3CoT benchmark (Chen et al., 2024c), SINKTRACK increases the accuracy of *Qwen2.5-VL-7B* by **22.8%** over `CoT`, and on the reading comprehension benchmark, SQuAD2.0 (Rajpurkar et al., 2018), it improves *Llama3.1-8B*'s accuracy by **21.6%**. Qualitative analysis reveals that SINKTRACK enhances the logical structure and detail focus in LLM's reasoning process (§3.3). We also explore the underlying working mechanism of SINKTRACK (§B).

## 2 METHODOLOGY

We first review the phenomena of *attention drift* and *attention sink* that motivate our method (§2.1). Then, we present the exploration undertaken to use *attention sink* as a dynamic information anchor, beginning with an initial `hard injection` (§2.2), progressing to a `soft injection` (§2.3), and leading to the development of SINKTRACK, our core *dual-track* mechanism (§2.4).

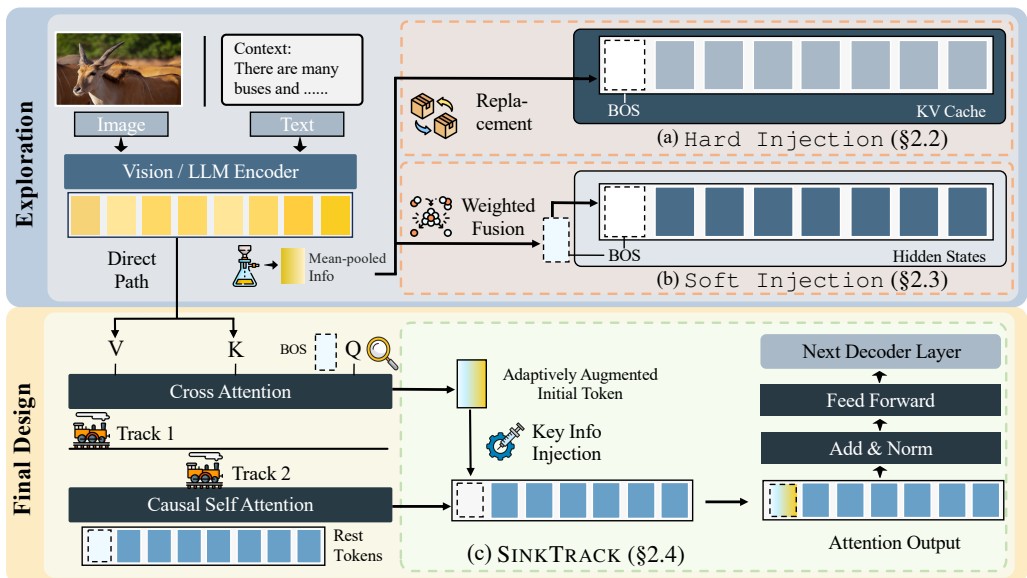

Figure 3: An overview of our methodology exploration.

## 2.1 ATTENTION PATTERNS OF LLMS

During auto-regressive generation, LLMs exhibit two fundamental and co-existing attention patterns that motivate our approach. The first is *attention drift* (Peysakhovich & Lerer, 2023; An et al., 2025; Liu et al., 2024a) (*cf.* Fig. 2), where LLMs' attention progressively concentrates on more recent tokens. As the generation proceeds, attention on the initial tokens, which contain the primary context (*e.g.*, instructions or image), dramatically decays. This progressive loss of focus on the original input is a primary cause of hallucination and context forgetting. The second pattern is *attention sink* (Xiao et al., 2024; Kaul et al., 2025). (*cf.* $1_{st}$ column in Fig. 2). In direct contrast to the general decay of attention on initial tokens, the very first token of the sequence (*i.e.*, ⟨BOS⟩) intrinsically and consistently maintains an exceptionally high degree of attention throughout the generation process. ⟨BOS⟩ is semantically sparse, yet it functions as LLMs' most stable point of attention.

These two patterns, one of decay and one of stability, operate in tandem. Surprisingly, while these phenomena have been studied independently, their synergistic potential has been overlooked. This presents a clear opportunity: rather than attempting to alter these inherent behaviors, this work, to our best knowledge, is the first to leverage their interplay. Our core idea is to repurpose the passive attention sink into an active mechanism to counteract drift. Detailed implementations are given next.

## 2.2 HARD INJECTION VIA DIRECT REPLACEMENT

This work targets **decoder-only LLM**, which is composed of $L$ Transformer decoder blocks stacked sequentially. The $l$-th block processes the sequence of hidden states from the preceding block, $\boldsymbol{H}^{(l-1)} \in \mathbb{R}^{n \times d}$, where $n$ is the sequence length and $d$ is the hidden dimension. Each decoder block contains two main sub-layers. The first is a multi-head attention (MHA) sub-layer:

$$\boldsymbol{M}^{(l)} = \text{LayerNorm}\big(\boldsymbol{H}^{(l-1)} + \text{MHA}(Q = \boldsymbol{H}^{(l-1)}, \ K = \boldsymbol{H}^{(l-1)}, \ V = \boldsymbol{H}^{(l-1)})\big). \quad (1)$$

The second sub-layer is a feed-forward network (FFN):

$$\boldsymbol{H}^{(l)} = \text{LayerNorm}\big(\boldsymbol{M}^{(l)} + \text{FFN}(\boldsymbol{M}^{(l)})\big). \quad (2)$$

**Implementation.** Based on the characteristic of attention sink, the most direct and simple method to implement our idea is to place information at the representation of ⟨BOS⟩. Therefore, we first attempt `hard injection` (*cf.* Fig. 3a), which directly manipulates the embedding of ⟨BOS⟩. Concretely, we directly replace the Value vector ($V$ in Eq. 1) corresponding to ⟨BOS⟩ in the KV cache with an external information vector $\boldsymbol{f}_{\text{info}}$. $\boldsymbol{f}_{\text{info}}$ is derived from the input context. For the textual task, it is the prompt embeddings; for the multi-modal task, it is the visual features from the MLLM's own vision encoder. It is then compressed via mean pooling to match the dimensionality.

**Results and Analysis.** Unfortunately, `hard injection` leads to a complete model collapse, yielding meaningless outputs. This indicates that LLM's internal computational pathways and

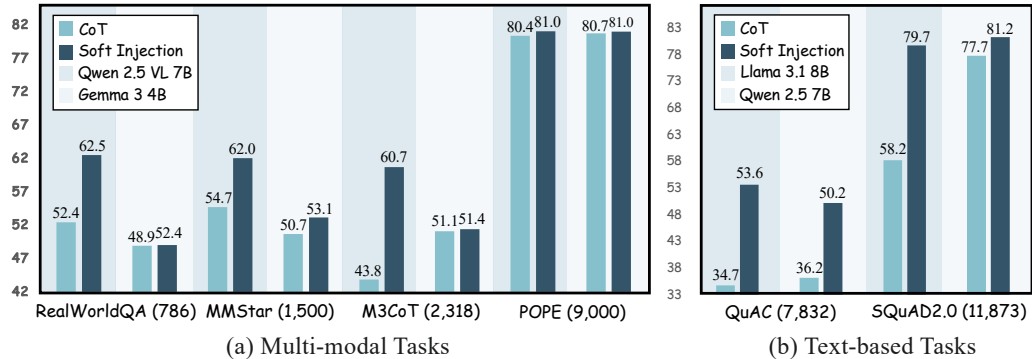

(a) Multi-modal Tasks (b) Text-based Tasks

Figure 4: `Soft injection` achieves considerable performance improvements (§2.3).

learned attention patterns are sensitive to direct manipulation. Abruptly overwriting values in the KV cache disrupts this finely-tuned structure, leading to a severe degradation. This finding establishes a core principle: any effective information injection must preserve the integrity of the LLM's native computational flow, motivating our development of the more nuanced `soft injection`.

## 2.3 SOFT INJECTION VIA STATIC FUSION

**Implementation.** In contrast to the disruptive replacement used in `hard injection`, `soft injection` fuses information into LLMs' computational flow. This is achieved by fusing a mean-pooled information vector with the original hidden state of ⟨BOS⟩ via weighted sum (*cf.* Fig. 3b):

$$\bar{\boldsymbol{H}}_0^{(l)} = \alpha \cdot \boldsymbol{H}_0^{(l)} + (1 - \alpha) \cdot \boldsymbol{f}_{\text{info}}, \tag{3}$$

where $\boldsymbol{H}_0^{(l)}$ is the original hidden state of the first token ⟨BOS⟩ at injection layer $l$, which corresponds to the output of the FFN sub-layer (Eq. 2). $\boldsymbol{f}_{\text{info}} \in \mathbb{R}^d$ is the mean-pooled information vector, $\alpha$ is the injection strength, and $\bar{\boldsymbol{H}}_0^{(l)}$ is the updated hidden state after `soft injection`. Such an injection strategy holds two main hyperparameters: the injection layers and the injection strength $\alpha$.

**Results and Analysis.** We evaluate `soft injection` in four multi-modal and two text-based tasks (§3.1). While soft injection achieves considerable gains (*cf.* Fig. 4) and validates our core premise, its static design reveals fundamental limitations. Its performance is contingent on a manually tuned hyperparameter (*i.e.*, injection strength $\alpha$), restricting its applicability. Furthermore, by uniformly aggregating all contextual information, it lacks the capacity to adaptively select the most salient content. This indiscriminate fusion can introduce noise and even degrade performance, highlighting the need for a dynamic mechanism that can adaptively query the source context.

## 2.4 SINKTRACK: DUAL-TRACK CROSS-ATTENTION MECHANISM

To overcome the limitations of soft injection, we introduce SINKTRACK, an adaptive mechanism centered on a dual-track attention design. SINK-TRACK achieves dynamic information infusion by separating the attention computation for ⟨BOS⟩ from that of all subsequent tokens (*cf.* Fig. 3c). This separation allows SINK-TRACK to inject context adaptively while preserving LLM's native computational flow.

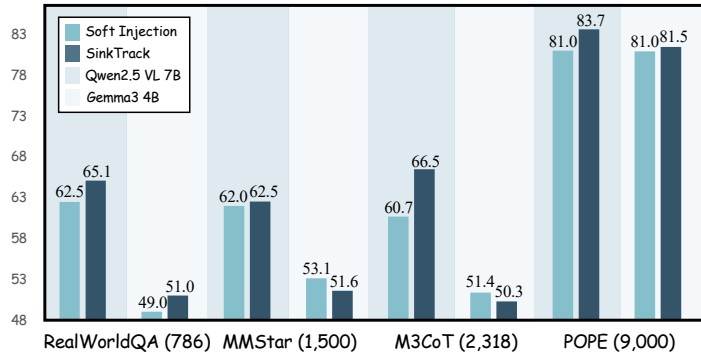

Figure 5: SINKTRACK further improves performance.

**Track 1: Adaptive Injection via Cross-Attention**. The first track performs adaptive injection exclusively for ⟨BOS⟩. This is achieved by modifying MHA (Eq. 1) at designated injection layers. Concretely, the hidden state of ⟨BOS⟩ at injection layer $l$, $\boldsymbol{H}_0^{(l)}$, serves as the Query ($Q$), while the Key ($K$) and Value ($V$) are now derived from the external, mean-pooled information vector ($\boldsymbol{f}_{\text{info}}$).

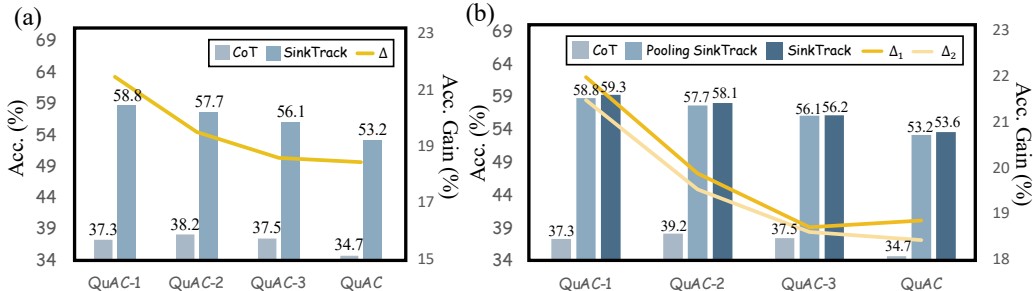

Figure 6: Effect of mean-pooling in information injection (Eq. 4). (a) Using the mean-pooled information vector; (b) using the original (unpooled) one.

This is implemented by cross-attention:

$$\bar{\boldsymbol{H}}_0^{(l)} = \text{MHA}(Q = \boldsymbol{H}_0^{(l)}, \ K = \boldsymbol{f}_{\text{info}}, \ V = \boldsymbol{f}_{\text{info}}), \tag{4}$$

where $\bar{\boldsymbol{H}}_0^{(l)}$ denotes the updated hidden state after cross-attention. While the cross-attention is designed for adaptive injection, one may wonder: how can the low-information query vector $\boldsymbol{H}_0^{(l)}$ retrieve information adaptively? We emphasize that this adaptive capability is a layer-wise information accumulation. At the initial injection layer, $\boldsymbol{H}_0^{(l)}$ aggregates a holistic representation of the global context from $\boldsymbol{f}_{\text{info}}$. This aggregation infuses $\boldsymbol{H}_0^{(l)}$ with context, thereby enabling it to formulate targeted queries in subsequent injection layers. This progression allows ⟨BOS⟩ to identify salient information dynamically, thus obviating the need for a manually tuned injection strength.

**Track 2: Preserving Native Computation via Self-Attention.** The second track ensures that LLM's computational flow remains undisturbed. All other tokens (*i.e.*, regular tokens) perform the standard causal self-attention computation, attending to the original, unmodified input sequence.

**Sequence Reassembly.** The two tracks operate in parallel within the same attention block. The final output is a concatenation of the updated ⟨BOS⟩ representation from Track 1 and the regular token representations from Track 2. This dual-track mechanism allows SINKTRACK to function as an adaptive context anchor, retrieving necessary information without compromising the foundational structure of LLMs. The superiority of this design is demonstrated by its consistent performance gains on long-context benchmarks like QuAC (*cf.* Fig. 5).

**The Effect of Mean-Pooling.** Despite outperforming CoT, we observe that SINKTRACK's performance gains on the long-dialogue QuAC dataset diminish as the context length grows (*cf.* Fig.6a). We identify the cause as the mean-pooling operation, a step inherited from hard and soft injection, which inevitably causes information loss by compressing a variable-length sequence into a fixed-size vector – a problem that intensifies with longer contexts. However, the cross-attention mechanism in SINKTRACK offers a structural advantage, as it does not require sequence length alignment between the Query and the Key/Value. We therefore eliminate the lossy pooling step, instead using the complete, uncompressed sequence of contextual features as the source. This modification effectively mitigates the trend of diminishing gains and preserves performance on long-context tasks (*cf.* Fig.6b). The complete process is described in Alg. S1 to ease understanding.

## 3 EXPERIMENT

### 3.1 EXPERIMENTAL SETUP

**Datasets.** We select four multi-modal datasets and two text-based datasets to assess the performance of SINKTRACK for different modalities.

• **Multi-modal Datasets.** Four multi-modal datasets are used for evaluation: **RealWorldQA** (X.AI, 2024), a visual question answering (QA) dataset designed to assess model robustness in complex, real-world scenarios; **MMStar** (Chen et al., 2024b), a comprehensive benchmark for assessing multi-modal capabilities that encompasses 6 core abilities across 18 main tasks; **M3CoT** (Chen et al., 2024c), a task collection focused on multi-modal, multi-hop, complex chain-of-thought reasoning; and **POPE** (Li et al., 2023c), a benchmark specifically designed to quantify object-level hallucination in multi-modal models.

| Model | RealWorldQA (768) | | MMStar (1,500) | | M3CoT (2,318) | | POPE (9,000) | | Average | |
|---|---|---|---|---|---|---|---|---|---|---|
| | Acc | Macro-F1 | Acc | Macro-F1 | Acc | Macro-F1 | Acc | Macro-F1 | Acc | Macro-F1 |
| *Qwen2.5-VL-3B-Instruct (Bai et al., 2025)* | | | | | | | | | | |
| Direct | $31.15_{\pm1.63}$ | $23.25_{\pm4.36}$ | $24.89_{\pm1.65}$ | $39.54_{\pm1.90}$ | $33.43_{\pm0.54}$ | $44.20_{\pm3.08}$ | $53.23_{\pm0.24}$ | $69.48_{\pm0.20}$ | 35.68 | 44.12 |
| CoT | $35.69_{\pm1.61}$ | $26.46_{\pm4.71}$ | $35.24_{\pm0.83}$ | $50.68_{\pm0.95}$ | $20.69_{\pm0.51}$ | $32.04_{\pm0.62}$ | $64.58_{\pm0.22}$ | $78.10_{\pm0.16}$ | 39.05 | 46.82 |
| SINKTRACK | $\mathbf{48.28}_{\pm0.40}$ | $\mathbf{28.42}_{\pm0.22}$ | $\mathbf{47.24}_{\pm0.40}$ | $\mathbf{61.38}_{\pm0.44}$ | $\mathbf{48.25}_{\pm0.09}$ | $\mathbf{58.73}_{\pm1.14}$ | $\mathbf{77.69}_{\pm0.12}$ | $\mathbf{87.44}_{\pm0.07}$ | **55.37** | **58.99** |
| *Gemma3-4B-Instruct (Team et al., 2025)* | | | | | | | | | | |
| Direct | $51.33_{\pm0.06}$ | $\mathbf{48.43}_{\pm1.10}$ | $35.47_{\pm0.20}$ | $52.17_{\pm0.12}$ | $27.71_{\pm0.20}$ | $33.81_{\pm0.62}$ | $83.98_{\pm0.04}$ | $91.30_{\pm0.02}$ | 49.62 | 56.43 |
| CoT | $49.41_{\pm1.02}$ | $44.53_{\pm5.27}$ | $51.36_{\pm1.53}$ | $67.52_{\pm1.32}$ | $50.96_{\pm0.32}$ | $\mathbf{59.71}_{\pm1.03}$ | $82.01_{\pm0.38}$ | $90.04_{\pm0.24}$ | 58.44 | 65.45 |
| SINKTRACK | $\mathbf{52.46}_{\pm0.43}$ | $47.39_{\pm1.59}$ | $\mathbf{53.04}_{\pm0.53}$ | $\mathbf{69.11}_{\pm0.48}$ | $\mathbf{51.01}_{\pm0.78}$ | $59.18_{\pm1.20}$ | $\mathbf{84.33}_{\pm0.02}$ | $\mathbf{91.50}_{\pm0.01}$ | **60.21** | **66.80** |
| *Qwen2.5-VL-7B-Instruct (Bai et al., 2025)* | | | | | | | | | | |
| Direct | $47.49_{\pm0.22}$ | $39.93_{\pm4.33}$ | $37.29_{\pm0.80}$ | $54.24_{\pm0.95}$ | $39.20_{\pm0.29}$ | $51.01_{\pm0.55}$ | $78.21_{\pm0.27}$ | $87.15_{\pm0.20}$ | 50.55 | 58.08 |
| CoT | $57.86_{\pm1.29}$ | $43.00_{\pm4.14}$ | $55.84_{\pm0.74}$ | $71.29_{\pm0.73}$ | $44.11_{\pm0.56}$ | $57.96_{\pm0.96}$ | $83.65_{\pm0.08}$ | $90.63_{\pm0.06}$ | 60.37 | 65.72 |
| SINKTRACK | $\mathbf{65.49}_{\pm0.37}$ | $\mathbf{52.81}_{\pm0.68}$ | $\mathbf{63.78}_{\pm0.19}$ | $\mathbf{77.86}_{\pm0.17}$ | $\mathbf{66.94}_{\pm0.21}$ | $\mathbf{79.13}_{\pm0.23}$ | $\mathbf{85.47}_{\pm0.02}$ | $\mathbf{91.67}_{\pm0.01}$ | **70.42** | **75.37** |
| *Gemma3-12B-Instruct (Team et al., 2025)* | | | | | | | | | | |
| Direct | $57.34_{\pm0.44}$ | $50.96_{\pm3.74}$ | $44.58_{\pm0.55}$ | $60.89_{\pm0.59}$ | $41.90_{\pm0.40}$ | $51.46_{\pm0.87}$ | $84.59_{\pm0.07}$ | $91.65_{\pm0.04}$ | 57.10 | 63.74 |
| CoT | $57.69_{\pm0.25}$ | $49.92_{\pm2.00}$ | $60.56_{\pm0.91}$ | $75.09_{\pm0.67}$ | $60.32_{\pm0.30}$ | $\mathbf{72.24}_{\pm0.83}$ | $84.53_{\pm0.33}$ | $91.61_{\pm0.19}$ | 65.78 | 72.22 |
| SINKTRACK | $\mathbf{57.91}_{\pm0.33}$ | $\mathbf{52.54}_{\pm0.77}$ | $\mathbf{61.60}_{\pm0.46}$ | $\mathbf{75.81}_{\pm0.51}$ | $\mathbf{60.57}_{\pm0.12}$ | $70.97_{\pm0.67}$ | $\mathbf{85.40}_{\pm0.13}$ | $\mathbf{92.13}_{\pm0.05}$ | **66.37** | **72.86** |

Table 1: Performance of MLLMs on multi-modal tasks in terms of Accuracy (%) and Macro-F1. The **bold** indicates the best performance for the base LLM.

• **Text-based Datasets.** Two text-based datasets are used for evaluation: **QuAC** (Choi et al., 2018), a conversational QA benchmark, to measure the model's ability to handle long-context dependencies. Its structure, where questions build upon the dialogue history, allows us to analyze performance across progressive dialogue turns (QuAC-1, -2, -3) as the context lengthens, thereby assessing the mitigation of context forgetting; and **SQuAD2.0** (Rajpurkar et al., 2018), another QA benchmark to test LLMs' ability to abstain from answering when information is absent in the context, providing a direct measure of its capacity to avoid generating fabricated content.

**Base LLMs.** We validate the effectiveness and generalizability of SINKTRACK across mainstream, open-source LLMs and MLLMs. Specifically, *Qwen2.5* (Qwen et al., 2025), *MiniCPM3* (Hu et al., 2024), and *Llama3.1* (Grattafiori et al., 2024) are used for text-based tasks, while *Qwen2.5-VL* (Bai et al., 2025) and *Gemma3* (Team et al., 2025) are used for multi-modal tasks.

**Competitors.** Since SINKTRACK is training-free and plug-and-play, `Direct` queries with LLMs, and Chain-of-Thought (Wei et al., 2022) (`CoT`) are used as our primary competitors. Specifically, CoT guides the model to perform step-by-step reasoning by including instructions, "Let's think step by step", in the prompt, without applying any additional information injection mechanism. This comparison allows for a clear isolation of the performance gains attributable to SINKTRACK.

**Implementation Details.** We adopt an intermittent injection strategy for both `soft injection` and SINKTRACK. Specifically, the injection operation is applied at intervals of every 5 layers. This configuration is empirically determined based on our analysis (§3.4.1), where we observe that such intermittent injection yields better performance compared to dense injection.

## 3.2 QUANTITATIVE COMPARISON RESULTS

To ensure reliability and rule out the influence of generation randomness, we conduct all evaluations using three distinct random seeds. Consequently, we report the performance as the mean ± variance across these independent runs. The quantitative results are presented in Table 1 for multi-modal tasks and Table 2 for text-based tasks. The findings show that SINKTRACK consistently outperforms both `Direct` and `CoT` by a clear margin across all tested LLMs, modalities, and benchmarks.

The results demonstrate that SINKTRACK consistently surpasses both `Direct` and `CoT` baselines across all tested LLMs (including Qwen2.5-VL and Gemma3) and benchmarks. Crucially, the low variance observed in SINKTRACK's performance indicates that anchoring the context via the attention sink not only improves accuracy but also enhances the stability of the model's reasoning process compared to standard prompting methods.

**Multi-modal Tasks.** On benchmarks designed to assess complex reasoning and object-level hallucination, SINKTRACK shows substantial gains. For instance, on M3CoT, SINKTRACK boosts the accuracy of *Qwen2.5-VL-7B* by **+22.8%** over `CoT`. On POPE, a benchmark specifically designed for hallucination detection, SINKTRACK consistently improves both accuracy and Macro-F1 scores

| Model | QuAC-1 (1,000) | | QuAC-2 (2,000) | | QuAC-3 (3,000) | | QuAC (7,354) | | SQuAD2.0 (11,873) | |
|---|---|---|---|---|---|---|---|---|---|---|
| | Acc | Macro-F1 | Acc | Macro-F1 | Acc | Macro-F1 | Acc | Macro-F1 | Acc | Macro-F1 |
| *MiniCPM3-4B (Hu et al., 2024)* | | | | | | | | | | |
| Direct | $52.82_{\pm0.62}$ | $52.79_{\pm0.40}$ | $49.97_{\pm0.34}$ | $50.94_{\pm0.23}$ | $48.13_{\pm0.36}$ | $49.53_{\pm0.23}$ | $47.17_{\pm0.19}$ | $48.07_{\pm0.05}$ | $62.93_{\pm0.15}$ | **$38.61_{\pm0.08}$** |
| CoT | $42.69_{\pm0.94}$ | $45.45_{\pm0.71}$ | $41.03_{\pm0.33}$ | $43.35_{\pm0.73}$ | $39.39_{\pm0.27}$ | $41.19_{\pm0.74}$ | $37.68_{\pm0.18}$ | $36.50_{\pm0.20}$ | **$63.63_{\pm0.11}$** | $23.73_{\pm0.04}$ |
| SINKTRACK | **$53.57_{\pm0.41}$** | **$52.82_{\pm0.43}$** | **$50.12_{\pm0.26}$** | **$50.98_{\pm0.25}$** | **$48.53_{\pm0.15}$** | **$49.75_{\pm0.21}$** | **$47.29_{\pm0.27}$** | **$48.08_{\pm0.30}$** | $62.74_{\pm0.04}$ | $36.83_{\pm0.09}$ |
| *Qwen2.5-7B-Instruct (Qwen et al., 2025)* | | | | | | | | | | |
| Direct | $52.57_{\pm0.39}$ | $47.06_{\pm0.20}$ | $52.00_{\pm0.20}$ | $48.06_{\pm0.10}$ | $51.26_{\pm0.22}$ | $47.64_{\pm0.15}$ | $49.47_{\pm0.07}$ | $46.71_{\pm0.15}$ | $80.97_{\pm0.13}$ | **$33.39_{\pm0.06}$** |
| CoT | $38.47_{\pm1.55}$ | $42.11_{\pm0.90}$ | $37.30_{\pm1.16}$ | $39.42_{\pm0.76}$ | $36.34_{\pm0.68}$ | $37.92_{\pm0.22}$ | $36.48_{\pm0.29}$ | $36.58_{\pm0.09}$ | $77.39_{\pm0.14}$ | $28.74_{\pm0.06}$ |
| SINKTRACK | $52.93_{\pm0.25}$ | $47.52_{\pm0.21}$ | $52.98_{\pm0.10}$ | **$48.08_{\pm0.07}$** | **$52.01_{\pm0.30}$** | $47.77_{\pm0.11}$ | **$50.25_{\pm0.21}$** | $46.73_{\pm0.15}$ | **$81.04_{\pm0.09}$** | $29.63_{\pm0.05}$ |
| *Llama3.1-8B-Instruct (Grattafiori et al., 2024)* | | | | | | | | | | |
| Direct | $54.10_{\pm0.02}$ | $48.87_{\pm0.14}$ | $54.25_{\pm0.08}$ | $51.31_{\pm0.09}$ | $53.70_{\pm0.04}$ | $51.20_{\pm0.13}$ | $52.45_{\pm0.07}$ | $50.66_{\pm0.15}$ | $78.69_{\pm0.02}$ | $24.95_{\pm0.04}$ |
| CoT | $54.90_{\pm0.12}$ | $46.09_{\pm0.09}$ | $51.75_{\pm0.24}$ | $38.21_{\pm0.11}$ | $49.13_{\pm0.27}$ | $34.47_{\pm0.13}$ | $46.95_{\pm0.29}$ | $28.74_{\pm0.05}$ | $58.19_{\pm0.31}$ | **$28.86_{\pm0.07}$** |
| SINKTRACK | **$59.40_{\pm0.10}$** | **$54.05_{\pm0.04}$** | **$58.05_{\pm0.05}$** | **$54.25_{\pm0.08}$** | **$56.20_{\pm0.09}$** | **$53.45_{\pm0.11}$** | **$53.51_{\pm0.06}$** | **$51.56_{\pm0.09}$** | $79.83_{\pm0.07}$ | $25.72_{\pm0.13}$ |

Table 2: Performance of LLMs on text-based tasks in terms of Accuracy (%) and Macro-F1. The **bold** indicates the best performance for the base LLM.

(*e.g.*, **85.5%** *vs* 83.7% accuracy and **91.8%** *vs* 90.6% F1 score compared to CoT), confirming its effectiveness in mitigating the generation of non-existent objects.

**Text-based Tasks.** We use long-context conversational QA to specifically measure the mitigation of context forgetting. As the conversational context lengthens from QuAC-1 to QuAC, the performance of baseline methods degrades due to attention drift. In contrast, SINKTRACK maintains a strong and often widening advantage. The most significant gains are seen with Llama3.1-8B, where on the longest-context QuAC setting (7,354 questions), SINKTRACK achieves an accuracy of **56.2%**, a **7.1%** increase over CoT. On SQuAD2.0, which tests the LLM's ability to abstain from answering unanswerable questions, SINKTRACK also yields consistent improvements.

**Analysis of CoT Performance.** It is worth noting that CoT sometimes underperforms Direct Prompting, an observation consistent with existing studies. This performance degradation is highly task-dependent. In long-context understanding tasks, such as summarization and QA, CoT can even degrade performance (Liu et al., 2025; Wang & Sun, 2025). Furthermore, while CoT excels in complex reasoning (Sprague et al., 2025) like M3CoT, in non-reasoning tasks, its extended chains exacerbate attention drift and raise the risk of error accumulation, where a single misstep derails the final answer (Shen et al., 2025; Liu et al., 2025).

Taking together, these results validate that SINKTRACK is a broadly effective and generalizable enhancement. It directly addresses both hallucination and context forgetting across different modalities, tasks, and model architectures without requiring any model training.

## 3.3 QUALITATIVE COMPARISON RESULTS

We present qualitative comparisons to illustrate how SINKTRACK improves LLM's reasoning. In cases where CoT fails (Fig. 7, left), SINKTRACK succeeds by grounding its analysis in specific contextual details (*e.g.*, skis and poles) rather than ambiguous cues. This advantage also extends to failure cases where both methods produce an incorrect answer due to knowledge deficits (Fig. 7, right). Even then, SINKTRACK's reasoning chain remains internally consistent, whereas CoT exhibits context forgetting, where its final conclusion contradicts its own intermediate steps. This indicates that SINKTRACK enhances the generation process by enforcing closer adherence to the initial context and maintaining logical integrity, even when LLM's underlying knowledge is flawed.

## 3.4 ANALYSIS

### 3.4.1 INJECTION STRATEGIES

To study the effects of two main hyperparameters: the injection layers and the injection strength (§2.3), we analyze the performance of various soft injection using *Qwen2.5-VL-7B-Instruct* on M3CoT. The results in Table 3 highlight two critical findings: **i)** Injection Frequency. Intermittent injection (Inject Every 5 Layers) is more effective than continuous injection (Inject Every Layer), with a 9.0% absolute accuracy improvement. This suggests that overly frequent intervention might disrupt LLM's native computational flow to some extent. **ii)** Injec-

| Method | Acc (%) |
|---|---|
| CoT (Baseline) | 43.83 |
| Inject Every Layer | 51.02 |
| Inject Every 5 Layers | 60.04 |
| Intermittent & Increasing | 59.87 |
| **Intermittent & Decaying** | **60.69** |

Table 3: Performance of different soft injection variants(§2.3).

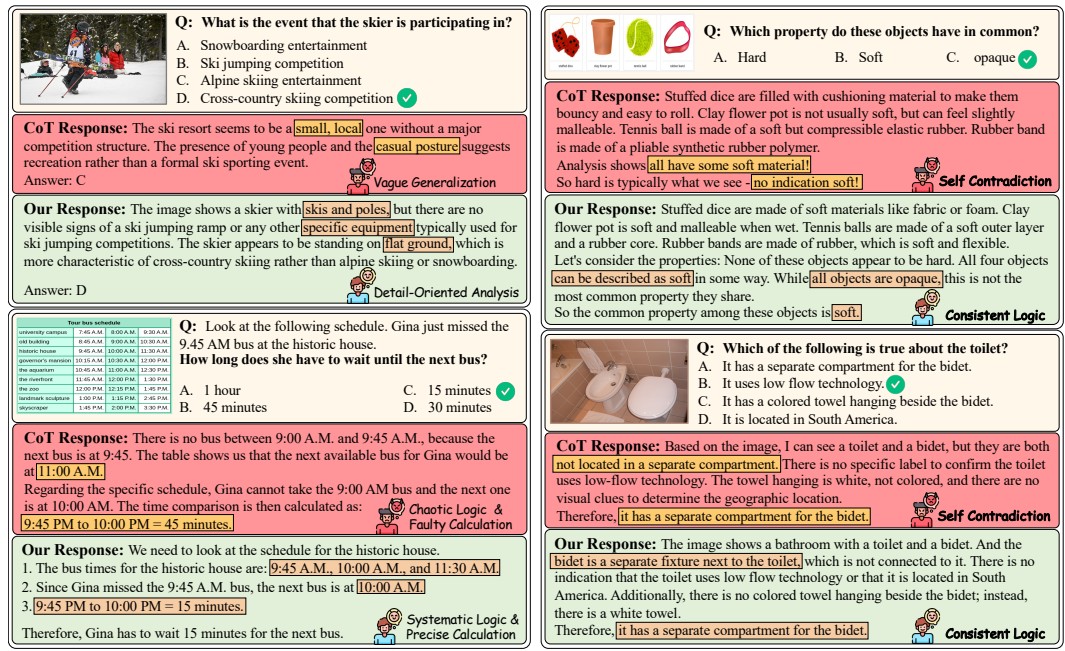

Figure 7: Qualitative comparison between CoT and SINKTRACK.

tion Strength. A decaying schedule, which applies stronger injection in earlier layers, yields better performance (60.7% *vs* 59.9%). This indicates that early contextual anchoring is more beneficial than later-stage interventions. In addition, the sensitivity to injection strength motivates us to develop an adaptive mechanism (§2.4).

### 3.4.2 HIERARCHICAL ATTENTION PATTERN

To analyze the structural impact of our injection, we quantify its effect on LLM's attention pattern. Concretely, we compute the Spearman's rank correlation between ⟨BOS⟩'s attention weight vectors across all layers, before and after applying SINKTRACK. The analysis reveals a near-perfect positive correlation ($\rho = 0.9985, p < 2.22 \times 10^{-17}$), as detailed in Appendix §B.1. This high correlation demonstrates that SINKTRACK preserves the relative importance of attention across different layers. Rather than disrupting LLM's pre-trained attention hierarchy, SINKTRACK enriches the information content of ⟨BOS⟩ while maintaining the integrity of the original computational flow.

### 3.4.3 INFORMATION FLOW

We further quantify the underlying working mechanism of SINKTRACK as an information anchor by analyzing its vertical and information flows.

**Vertical Flow: Encoding via Value Vectors.** The output of the attention mechanism is an aggregation of value vectors weighted by attention scores:

$$O_t = \sum_{j=1}^{t} \alpha_{t,j} \cdot V_j, \tag{5}$$

where $\alpha_{t,j}$ represents the attention weight and $V_j$ denotes the value vector of the $j$-th token. Consequently, the information capacity of the ⟨BOS⟩ anchor can be quantified by calculating the L1 Norm of its value vector. As illustrated in Fig. 8(a), a significant difference value (yellow line) is observed between the original state (Before) and the injected state (After). This variation serves as a proxy for *Information Gain*. The distinct divergence in L1 Norm across layers confirms that the anchor information is successfully absorbed and encoded into the ⟨BOS⟩ representation, effectively transforming it from a standard placeholder into a semantic anchor.

**Horizontal Flow: Sustained Attention Anchoring.** To evaluate how this injected information is propagated during generation, we analyze the attention distribution of newly generated tokens. Specifically, we track the attention weights that subsequent tokens assign to the ⟨BOS⟩ anchor

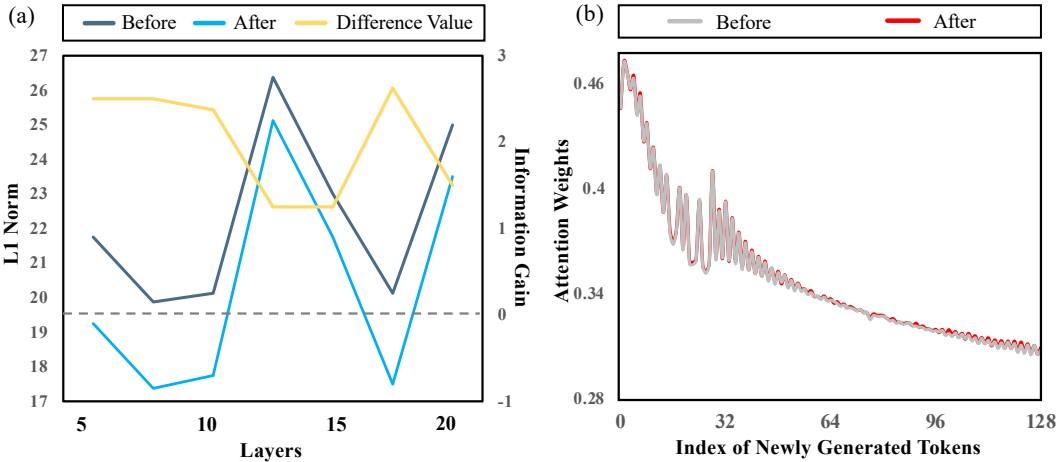

Figure 8: The illustration of information flow with and without SINKTRACK (§2.4).

($\alpha_{t,\text{BOS}}$). As shown in Fig. 8(b), the attention weights exhibit a natural decay but maintain a persistent magnitude, indicating that the generated sequence continuously attends to the ⟨BOS⟩ token. Crucially, the attention curves before and after injection (gray and red lines) overlap almost perfectly. This phenomenon demonstrates two key findings: **i**) SINKTRACK successfully establishes a long-term information bridge, as the anchor remains a focal point throughout generation; and **ii**) The injection is non-destructive, preserving the intrinsic attention patterns of the LLMs.

## 4 RELATED WORK

**Training-free Inference-time Enhancement for LLMs**. Developing plug-and-play enhancement methods to improve the capabilities of LLMs has received widespread attention from the community. These approaches can be broadly categorized into three main streams. The first category is *External Module Augmentation* (Wang et al., 2024; Ma et al., 2024; Feng et al., 2024; Chen et al., 2022). These methods couple an LLM with external components to address limitations such as outdated knowledge or the lack of specific skills, *e.g.*, retrieval-augmented (Wang et al., 2024; Feng et al., 2024; Yu et al., 2024a) and tool-augmented methods (Ma et al., 2024; Gou et al., 2024; Lu et al., 2023; Chen et al., 2022; Wei et al., 2022). The second category, *Internal Reasoning Process Enhancement* (Dhuliawala et al., 2024; Zhang et al., 2024c; Yao et al., 2023; Zhang et al., 2024a; 2025a), uses structured prompting strategies or adds reasoning steps to guide LLM, *e.g.*, decomposing complex problems into simpler sub-tasks (Khot et al., 2023; Chen et al., 2022; Wei et al., 2022; Ning et al., 2024), generating multiple candidate outputs with verification (Gou et al., 2024; Wang et al., 2023; Zhang et al., 2024b;a), and using non-linear reasoning structures to organize complex inference paths (Yao et al., 2023; Roy & Roth, 2015; Besta et al., 2024; Wei et al., 2022). The third category is *Internal Representation Intervention* (Li et al., 2023a; Zhu et al., 2025; Han et al., 2025; Turner et al., 2023; Liu et al., 2024c), where methods directly modify the LLM's internal states during inference to guide its behavior, *e.g.*, adding a targeted offset to LLM's hidden states (Li et al., 2023a; Subramani et al., 2022; Turner et al., 2023) and calibrating LLM's attention (Zhu et al., 2025; Yu et al., 2024b; Han et al., 2025; Lu et al., 2021). Despite impressive, these methods often require repetitive intervention, introducing considerable computational cost at each generation step.

Unlike methods that apply continuous modifications during the decoding loop, SINKTRACK achieves training-free enhancement through a one-time intervention prior to inference. The injection of key information into the representation of ⟨BOS⟩ is a one-off operation performed before generation begins. Once this information anchor is established, the subsequent auto-regressive process is identical to the standard decoding procedure, requiring no extra computational steps.

**Attention Sink** refers to the empirical observation that the initial token of a sequence (*i.e.*, ⟨BOS⟩) consistently receives a high degree of attention during auto-regressive generation. This phenomenon is considered an emergent property of Transformer architectures (Xiao et al., 2024; Kaul et al., 2025; Gu et al., 2025). Proposed causes include it being an artifact of the Softmax function within the attention mechanism (Barbero et al., 2025) or a consequence of the token's stable structural position in the sequence (Yu et al., 2024b; Ruscio et al., 2025; Cancedda, 2024). Existing efforts use the attention sink in two main ways. One line of work uses the initial token's stability for

tasks like KV cache management (Zhang et al., 2023) and key token protection (Liu et al., 2024b). Another research direction focuses on enhancing LLM's performance by modifying or calibrating the attention mechanism to synergize with this structural anchor (Han et al., 2025; Gu et al., 2025).

This work explores a new application branch for attention sink: repurposing it as an active mechanism for context anchoring through information injection. We do not alter the attention weights themselves; instead, we modify the representation of ⟨BOS⟩ to form a high-information anchor that provides LLMs with high-integrity context. This shift from indirect attention calibration to direct information supply reveals a new direct to address the issues of hallucination and context forgetting.

## 5 CONCLUSION

This work mitigates two long-standing issues (*i.e.*, hallucination and context forgetting) in decoder-only LLMs by exploiting the synergistic relationship between attention drift and attention sink. We propose SINKTRACK, an efficient, plug-and-play method that transforms the passive attention sink into an active context anchor. Through an adaptive cross-attention mechanism, SINKTRACK injects key contextual information into the first token's representation, thereby counteracting the information decay caused by attention drift during generation. Consistent performance improvements across diverse textual and multi-modal benchmarks validate its effectiveness and generalizability, and our analysis of its information delivery mechanism further substantiates the principles of SINKTRACK.

## ETHICS STATEMENT

In this study, we propose SINKTRACK, a training-free mechanism designed to enhance the reliability and reasoning capabilities of LLMs. We acknowledge that any technique that improves the fundamental capabilities of LLMs – such as increasing their coherence or reducing factual errors – carries a dual-use potential. While our work is intended for benevolent purposes, more capable and robust models could theoretically be exploited by malicious actors to generate more persuasive and harder-to-detect misinformation.

Our research is focused exclusively on the positive application of mitigating object hallucinations and improving long-context reasoning on established, public academic benchmarks. The datasets and tasks used in our experiments do not involve sensitive personal data, human subjects, or the generation of socially harmful content. Our methodology is transparent and designed to be analyzed and understood by the research community.

## REPRODUCIBILITY STATEMENT

All experiments are conducted using publicly available, open-source LLMs and MLLMs. Specific model checkpoints are obtained from the official Hugging Face Hub. The detailed formulation and algorithmic implementation of SINKTRACK are presented in §2. A comprehensive description of our experimental setup, including the specific datasets, evaluation metrics, and baselines used, is provided in §3. Any additional implementation details are further detailed in the Appendix. To facilitate full reproducibility, our complete source code, along with the scripts required to run all experiments and reproduce the results reported in this paper, is available at GitHub.

## ACKNOWLEDGEMENT

This work was supported by National Science and Technology Major Project (No. 2023ZD0121300), National Natural Science Foundation of China (No. 62372405, T25B2005), Fundamental Research Funds for the Central Universities (226-2025-00057), Jiangsu Provincial Scientific Research Center of Applied Mathematics (No. BK20233002), and CCF-Tencent Open Fund.

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

## SUMMARY OF THE APPENDIX

- §A details the pseudo-code implementation of SINKTRACK.

- §B provides in-depth analyses concerning the stability of the hierarchical attention pattern, resistance to attention drift, and computational overhead.

- §C discusses the limitations of the current approach.

## A  PSEUDO-CODE FOR SINKTRACK

To provide a clear description of SINKTRACK, this section details the pseudo-code for it. As discussed in the main paper, the core idea is to create two parallel computational tracks within the attention module. *Track 1 (Injection Track)* uses cross-attention to inject external information into the initial token. *Track 2 (Original Track)* performs standard causal self-attention for the rest of the tokens to maintain the integrity of the pre-trained sequence structure. The complete process, including the conditional check for applying the injection, is described in Algo. S1.

---

**Algorithm S1** Pseudo-code for the Dual-Track Cross-Attention Mechanism.

```
"""
h_ori: hidden states of the original sequence (L_ori × D_h)
h_info: hidden states of the external information (L_info × D_h)
cfg: configuration for injection rules
"""
def Hybrid_Attention(h_ori, h_info, cfg):
    #--- 1. Check if conditions for injection are met ---
    do_inject = Check_Conditions(cfg, h_info)

    if do_inject:
        #--- PATH A: HYBRID ATTENTION FOR INJECTION ---
        # 2a. Project inputs to Q, K, V spaces
        Q_ori, K_ori, V_ori = Project_QKV(h_ori)
        K_info, V_info = Project_KV(h_info)

        # 2b. Split the original Query for dual attention tracks
        q_first, Q_rest = Split_Original_Vector(Q_ori)

        # 3. Execute dual-track attention
        # Track 1: INJECTION via Cross-Attention
        o_first = Cross_Attention(Q=q_first, K=K_info, V=V_info)

        # Track 2: CAUSAL SELF-ATTENTION
        m = Create_Causal_Mask(Q_rest)
        o_rest = Causal_Self_Attention(Q=Q_rest, K=K_ori, V=V_ori, mask=m)

        # 4. Merge outputs from both tracks
        o_combined = Concatenate(o_first, o_rest)

    else
        #--- PATH B: STANDARD CAUSAL SELF-ATTENTION ---
        o_combined = Standard_Causal_Attention(h_ori)

    # 5. Apply final output projection
    h_out = Output_Projection(o_combined)

    return h_out
```

---

## B  IN-DEPTH ANALYSIS: MECHANISM AND EFFICIENCY

This section provides a deeper analysis of the underlying mechanism of SINKTRACK and its computational implications.

### B.1  STABILITY OF HIERARCHICAL ATTENTION PATTERN

To analyze the structural impact of our injection, we quantify its effect on the LLM's hierarchical attention pattern. Concretely, we compute the Spearman's rank correlation between the $\langle$BOS$\rangle$'s attention weight vectors across all layers, before and after applying SINKTRACK. As shown in Fig. S1, the analysis reveals a near-perfect positive correlation ($\rho = 0.9985, p < 2.22 \times 10^{-17}$).

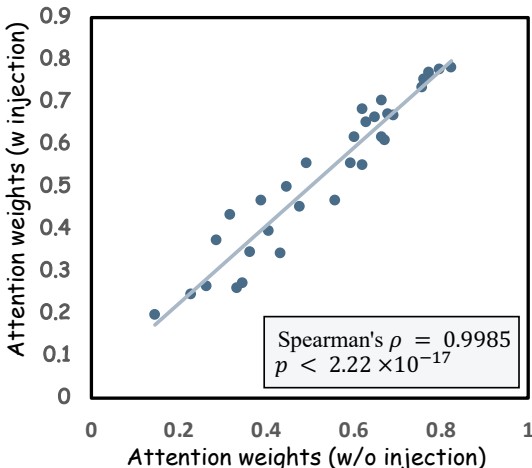

Figure S1: Comparison between the $\langle BOS \rangle$'s attention weight vectors before and after applying SINKTRACK.

This high correlation demonstrates that SINKTRACK preserves the relative importance of attention across different layers. Rather than disrupting the LLM's pre-trained attention hierarchy pattern, SINKTRACK enriches the information content of the initial token while maintaining the integrity of the original computational flow. This finding provides a quantitative explanation for why SINKTRACK can enhance performance without causing the model collapse observed in `hard injection`.

## B.2 VERIFICATION OF SINKTRACK'S STABILITY

To further verify that SINKTRACK remains effective over long generation sequences, we conduct a "Drift Test" on Llama3.1-8B. We feed it an input context of 8k+ tokens and track the attention weights allocated to the $\langle BOS \rangle$ token by subsequently generated tokens (steps 1 to 1024).

As presented in Table S1, the attention weight to $\langle BOS \rangle$ remains dominant throughout the generation process. For instance, at generated token index 1024 (sequence position 9106), the attention to $\langle BOS \rangle$ is 0.582, which is approximately $14\times$ higher than the maximum attention paid to any other token. This confirms that SINKTRACK creates a robust attention sink that is immune to attention drift, ensuring long-term context retention.

Table S1: SinkTrack Drift Test on Llama3.1-8B. The attention to $\langle BOS \rangle$ consistently dominates other tokens even after generating 1024 new tokens.

| Gen. Idx | Seq. Pos | Attn to $\langle BOS \rangle$ | Max Attn (Others) | Ratio |
|---|---|---|---|---|
| 1 | 8083 | 0.561 | 0.057 | 9.87x |
| 64 | 8146 | 0.570 | 0.048 | 11.97x |
| 128 | 8210 | 0.612 | 0.054 | 11.25x |
| 256 | 8338 | 0.560 | 0.050 | 11.24x |
| 512 | 8594 | 0.558 | 0.037 | 15.17x |
| 1024 | 9106 | 0.582 | 0.042 | 13.96x |

## B.3 COMPUTATIONAL OVERHEAD

We measure the prefill stage latency on an NVIDIA GeForce RTX 4090 to evaluate the computational cost of SINKTRACK. The results, averaged over 500 samples, are presented in Table S2. Specifically, Llama3.1-8B shows a latency increase from 35.90 ms *(w/o Injection)* to 36.66 ms *(w/ Injection)*, representing a marginal overhead of only +0.76 ms. Similarly, Qwen2.5-VL-7B shows an increase of +3.13 ms. These findings confirm that SINKTRACK introduces negligible latency, as the core intervention is a one-off operation performed prior to the auto-regressive generation.

Table S2: Prefill stage latency measurement on NVIDIA GeForce RTX 4090 (averaged over 500 samples). SINKTRACK introduces negligible computational overhead.

| Model | w/o Injection | w/ Injection | Overhead |
|---|---|---|---|
| Llama3.1-8B | 35.90 ms | 36.66 ms | +0.76 ms |
| Qwen2.5-VL-7B | 107.13 ms | 110.26 ms | +3.13 ms |

## C LIMITATION

This work proposes and validates SINKTRACK representing a solid step towards mitigating hallucination and context forgetting in LLMs. However, it is important to recognize several inherent limitations in our current approach that open avenues for future research. Specifically, SINKTRACK's core design anchors all contextual information to the representation of a single initial token. While our adaptive mechanism optimizes information retrieval, this setup introduces a potential information bottleneck, as a single fixed-dimension vector has a finite capacity for representation. This may not fully capture the nuanced informational needs of tasks requiring extremely long or dense contexts. For example, while effective, the diminishing performance gains we observe on the long-dialogue QuAC may be an early indicator of this bottleneck. This limitation suggests a need to explore future paradigms beyond a single anchor, such as distributed or dynamically positioned anchoring mechanisms.

