# OpenReview forum: "SinkTrack: Attention Sink based Context Anchoring for Large Language Models"
_ICLR.cc/2026/Conference — ICLR 2026 Poster_

### Official Review · Reviewer_YQZu · 2025-10-18

**Soundness:** 3
**Presentation:** 3
**Contribution:** 3
**Rating:** 6
**Confidence:** 3

**Summary:**

The paper proposes SINKTRACK, a training-free, dual-track cross-attention mechanism, solves the LLM problem of "forgetting" the original input by turning the naturally stable first token into an active information anchor that constantly and adaptively retrieves the most important details from the context.

**Strengths:**

1. The paper presents novel solutions to mitigate context forgetting and hallucination in LLMs by actively leveraging the model's intrinsic "attention sink" tendency towards the initial <bos> token.
2. The quality of the work is evident through the systematic exploration of context anchoring methods, such as soft injection and SINKTRACK, which effectively enhances context coherence. The empirical results on relevant datasets confirm the method’s efficacy and significance as a solid step forward in improving LLM reliability over long contexts.
3. This paper excels in clarity. The methodology is simple and well-motivated, and the overall organization of the paper is clear and easy to follow.

**Weaknesses:**

1. Sticking all the important context onto just the first token might create an information bottleneck for really long documents. The author needs to consider the impact of this.

2. The 'CoT' baseline is too simple, making the new method's performance improvements look bigger than they might actually be. The author needs to compare with more advanced methods.

3. It is critical to include experiments demonstrating that SINKTRACK does not hurt the model's general abilities, such as common sense reasoning and instruction following, in standard short-context benchmarks.

**Questions:**

1. Why is the $L_2$ norm used for Information Gain in Equation 5 instead of cosine similarity or another metric?

2. In "soft injection," does the "information vector" refer to the vector corresponding to the image information?

3. Does soft injection perform fusion on the $\langle\text{bos}\rangle$ hidden state before the key-value projection, and how is the $\alpha$ coefficient set?

4. Why does the CoT (Chain-of-Thought) method show a decrease in performance compared to the baseline in Table 2?

---

> ### Author Response · Authors · 2025-11-20
> **Response to Reviewer YQZu (1/2)**
>
> We thank reviewer YQZu for the valuable time and constructive feedback. We provide point-to-point response below.
>
> **Q1: Potential information bottleneck of a single-token anchor.**
>
> Thanks for your feedback. We acknowledge that compressing context into a single token presents an information bottleneck.
>
> However, surprisingly, our **extensive evaluation** demonstrate SinkTrack’s practical utility across a wide array of tasks. This includes complex **multi-hop reasoning** (M3CoT), **general capabilities** (MMStar, with 18 sub-tasks), **real-world VQA** (RealWorldQA), **hallucination detection** (POPE), **dialogue and context understanding** (QuAC, SQuAD v2).
>
> Furthermore, we now add a diverse suite of tasks in **LongBench [ref1]** (covering 6 major categories and 20 sub-tasks, including **single/multi-document QA, summarization, few-shot learning, and code completion, etc**.) with inputs **ranging from 5k to 15k** tokens in length. The results are shown in Q2. These results suggest that, in practice, SinkTrack allows \<BOS\> to effectively query and retrieve the most salient information, mitigating this bottleneck.
>
> Further modeling and addressing this potential bottleneck will be a top priority in our future work.
>
> **Q2: Limited baselines.**
>
> Good suggestion!
>
> To address this, we **expand our evaluation to include comparisons against multiple recent, open-source SOTA training-free methods**. For multi-modal tasks, we now compare against **VCD [ref4], OPERA [ref11],** **ICD [ref5], and SID [ref6]** on the POPE:
>
> - **VCD** mitigates reliance on language priors by contrasting output distributions from original versus distorted images.
> - **OPERA** counters the "over-trust" issue by applying a logit penalty and retrospection-allocation strategy during decoding, reducing over-reliance on summary tokens over image content.
> - **ICD** amplifies visual information by contrasting outputs generated from original versus disturbance-added instructions.
> - **SID** employs a self-introspection mechanism, leveraging the contrast between an initial draft and a self-critiqued revision to guide factual generation.
>
> | **Method** | **Accuracy (%)** | **F1-score** |
> | --- | --- | --- |
> | VCD [ref4] | 84.63 | 82.83 |
> | OPERA [ref11] | 84.67 | 83.01 |
> | ICD [ref5] | 84.71 | 82.94 |
> | SID [ref6] | 85.24 | 85.36 |
> | **SinkTrack (Ours)** | **86.52** | **86.05** |
>
> For text-modal tasks, we conduct **new experiments on LongBench** [ref1]. We compare against **InfLLM [ref2]** on LongBench. InfLLM stores distant contexts into additional memory units and employs an efficient mechanism to lookup token-relevant units for attention computation.
>
> | **Model (Llama3.1-8B)** | **0-4k** | **4-8k** | **8k+** | **Average** |
> | --- | --- | --- | --- | --- |
> | Direct | 39.58 | 40.54 | 43.11 | 41.08 |
> | CoT | 35.91 | 38.35 | 41.03 | 38.43 |
> | InfLLM [ref2] | 36.73 | 37.19 | 39.77 | 39.23 |
> | **SinkTrack (Ours)** | **40.43** | **41.66** | **44.32** | **42.14** |
>
> | **Model (Qwen2.5-7B)** | 0-4k | 4-8k | 8k+ | total |
> | --- | --- | --- | --- | --- |
> | Direct | 40.17 | 44.32 | 46.92 | 43.8 |
> | CoT | 33.42 | 38.05 | 40.03 | 37.17 |
> | InfLLM [ref2] | 35.7 | 38.62 | 38.84 | 37.72 |
> | **SinkTrack (Ours)** | **40.58** | **44.9** | **47.52** | **44.34** |
>
> The results show that SinkTrack consistently outperforms these advanced methods.
>
> It is worth noting that InfLLM was originally developed for Llama2 and Qwen1.5. To ensure a fair comparison on newer architectures, we implemented it as follows: For Llama3.1, we utilized the subsequent code extension **released by the authors**. For Qwen2.5, we analyzed the source code and confirmed that the interface compatibility allows for direct adaptation (e.g., updating the class invocation from QwenModel to Qwen2Model). Despite this valid adaptation, its performance remains suboptimal, as confirmed by **community feedback [ref3]** and our results. This further highlights the strong generalizability of SinkTrack.
>
> We will include the new results in the revision.

---

> ### Author Response · Authors · 2025-11-20
> **Response to Reviewer YQZu (2/2)**
>
> **Q3: Whether SinkTrack harms general model abilities.**
>
> Sorry for the misunderstanding. Our extensive experiments provide evidence that SinkTrack **DOES NOT harm, but rather enhances** the model's general abilities across a wide spectrum of tasks.
>
> Crucially, **SinkTrack is designed as a general-purpose enhancement**, compatible with diverse model architectures, tasks, and modalities, rather than being tailored to a specific niche. Our evaluation confirms this universality, covering **complex reasoning** (M3CoT), **general multi-modal capabilities** (MMStar), **dialogue and context understanding** (QuAC, SQuAD v2), and a diverse suite of tasks in **LongBench** (including **single/multi-document QA, summarization, few-shot learning, and code completion, etc.**).
> These results demonstrate that **equipping LLMs with SinkTrack consistently augments the LLM's general capabilities** across diverse tasks. This confirms that SinkTrack not only preserves but **improves** the LLM's performance.
>
> **Q4: Choice of L2 norm.**
>
> Good question！
>
> We choose the L2 norm for two primary reasons:
>
> 1. The L2 norm directly measures the magnitude of change in a token's hidden state, capturing the **strength of the perturbation**, unlike cosine similarity which only measures directional change.
> 2. It is more robust to cross-modal bias in MLLMs, as it is direction-agnostic and provides a stable measure of actual change, unaffected by initial **spatial differences** between text and image features.
>
> We will add a justification for this choice in the revision.
>
> **Q5&6: Implementation details of the soft injection.**
>
> Thanks for the insightful comment
>
> In 'soft injection', the 'information vector' is the mean-pooled vector from **image features** for multi-modal tasks. The fusion is performed on the \<BOS\> hidden state **after the FFN sub-layer and before it is used in the next layer's attention block**.
>
> Regarding the α coefficient, our ablation study (shown in Table 3) finds that a **decaying schedule** (linearly decreasing from **0.8 to 0.2**) yield the best performance, as stated in our paper (L391-395):
>
> > A decaying schedule, which applies stronger injection in earlier layers, yields better performance. This indicates that early contextual anchoring is more beneficial than later-stage interventions.
> >
>
> We will add a clearer description of these implementation details in the revision.
>
> **Q7: Performance decrease of the CoT baseline.**
>
> Thanks for your feedback.
>
> The observation that CoT can underperform Direct Prompting is a well-documented phenomenon in the field.
>
> For example, recent studies find that for long-context understanding tasks like document summarization and QA, CoT can even degrade performance [ref9-10]. Additionally, while CoT excels in complex reasoning [ref7], as shown in our M3CoT results, for non-reasoning tasks, generating long chains is counterproductive. It exacerbates long-context issues like attention drift and raises the risk of error accumulation, where a single misstep derails the final answer [ref8-9].
>
> We add a concise discussion to the revision to explain these results:
>
> > We observe that CoT sometimes underperforms Direct Prompting, an observation consistent with existing studies. This performance degradation is highly task-dependent. In long-context understanding tasks, such as summarization and QA, CoT can even disadvantages [ref9-10]. Furthermore, while CoT excels in complex reasoning [ref7] like M3CoT, in non-reasoning tasks, its extended chains exacerbate attention drift and raise the risk of error accumulation, where  a single misstep derails the final answer [ref8-9].
> >
>
> [ref1] LongBench: A Bilingual, Multitask Benchmark for Long Context Understanding, ACL 2024
>
> [ref2] InfLLM: Training-Free Long-Context Extrapolation for LLMs with an Efficient Context Memory, NeurIPS 2024
>
> [ref3] https://github.com/thunlp/InfLLM/issues/51
>
> [ref4] VCD: Mitigating Object Hallucinations in Large Vision-Language Models through Visual Contrastive Decoding, CVPR 2024
>
> [ref5] Mitigating Hallucinations in Large Vision-Language Models with Instruction Contrastive Decoding, ACL 2024
>
> [ref6] Self-Introspective Decoding: Alleviating Hallucinations for Large Vision-Language Models, ICLR 2025
>
> [ref7] To CoT or not to CoT? Chain-of-thought helps mainly on math and symbolic reasoning, ICLR 2025
>
> [ref8] Adaptive Chain-of-Thought Distillation Based on LLM Performance on Original Problems, Mathematics 2025
>
> [ref9] Mind Your Step (by Step): Chain-of-Thought can Reduce Performance on Tasks where Thinking Makes Humans Worse, arxiv 2024
>
> [ref10] Unable to Forget: Proactive Interference Reveals Working Memory Limits in LLMs Beyond Context Length, ICML 2025
>
> [ref11] OPERA: Alleviating Hallucination in Multi-Modal Large Language Models via Over-Trust Penalty and Retrospection-Allocation, CVPR 2024

---

> > ### Comment · Reviewer_YQZu · 2025-11-25
> >
> > Thank you for the response. Considering the feedback from other comments, I strongly suggest that the authors provide **all model output in evaluations** to substantiate the reported improvements. Otherwise, I will consider changing my score.

---

> > > ### Author Response · Authors · 2025-11-25
> > >
> > > Thanks again for your valuable suggestion! To ensure full reproducibility, we provide complete artifacts in our anonymous repository. We upload all code, inference results, and evaluation scripts so that every reported number can be recomputed from our official implementation. Pls see README for step-by-step instructions. If you encounter any issues or have any further concerns, we would be very happy to clarify details or provide additional materials.

---

> > > > ### Comment · Reviewer_YQZu · 2025-11-28
> > > >
> > > > I would like to thank the authors for their thorough rebuttal.
> > > >
> > > > The additional experiments and clarifications regarding reproducibility are appreciated and have resolved some of my initial questions.
> > > > However, my primary reservation remains regarding the theoretical understanding of the SINKTRACK method. The rebuttal successfully demonstrates its empirical strength, but the fundamental questions about its application limits and the nature of the potential information bottleneck are not yet fully explored.
> > > >
> > > > For this reason, I believe my current score remains appropriate. I would be happy to reconsider and raise my score if the paper could offer deeper insights into the mechanism of SINKTRACK, moving beyond the benchmark results to explain why it is effective.

---

### Official Review · Reviewer_85VA · 2025-10-18

**Soundness:** 2
**Presentation:** 3
**Contribution:** 2
**Rating:** 4
**Confidence:** 4

**Summary:**

This paper proposes a test-time method to mitigate reduce hallucination in MLLMs, and context forgetting on both LLMs and MLLMs. It uses soft injection to fuse information into LLM's computational flow.

**Strengths:**

1. The paper is generally well-written and easy to follow.
2. The authors conducted comprehensive experiments on multiple tasks and multiple LLMs to validate their idea.

**Weaknesses:**

1.	Methodological clarity:
Some details in the method section remain unclear. For example, it is not explicitly stated how f_{\text{info}} is obtained. From Figure 3, it appears that f_{\text{info}} results from mean pooling over the encoder outputs; however, the architecture details of the Vision/LLM encoder are missing. The authors also mention that this work focuses only on decoder-only LLMs, which further raises questions about how the encoder component is integrated. In addition, an analysis of the extra computational overhead introduced by this module would be helpful.
2.	Experimental setup:
In Table 2, it is unclear why Chain-of-Thought (CoT) prompting performs worse than Direct Prompting, particularly on text-only tasks. Moreover, the current baselines seem rather limited. Including stronger or more diverse baselines would better demonstrate the effectiveness and generalizability of the proposed method.
3. Definition ambiguity:
The distinction between vertical and horizontal information flow is not clearly justified. It is unclear whether this formulation is newly introduced by the authors or adapted from prior work. Providing clearer definitions and theoretical grounding would strengthen the analysis.
4. Potential information loss and generalizability:
The proposed approach compresses all information into a single token, which may still lead to information loss. Although the reported results on the two tasks are promising, the generalizability of this information compression strategy remains uncertain. Further empirical validation on broader or more diverse tasks would make the claims more convincing.

**Questions:**

see weakness

---

> ### Author Response · Authors · 2025-11-20
> **Response to Reviewer 85VA (1/3)**
>
> We thank reviewer 85VA for the valuable time and constructive feedback. We provide point-to-point response below.
>
> **Q1: Acquisition of $f_{info}$ and encoder integration.**
>
> Sorry for the misunderstanding.
>
> For multi-modal tasks,  $f_{info}$ is the image embedding obtained from the MLLM's **native vision encoder**. For text-modal tasks, it is the text embedding extracted by the LLM itself. Specifically, this is the output of the LLM's native **Embedding Layer** during the prefill stage. Note that, $f_{info}$ is reused from the model's standard forward pass and requires **no extra modules (encoders).**
>
> Our work aligns with the dominant paradigm in modern LLMs. The vast majority of mainstream LLMs (like Llama and Qwen series) are **Decoder-only** architectures. While MLLMs possess a vision encoder (which we utilize), we do not consider Encoder-Decoder models (like T5) or Encoder-only models (like BERT).
>
> We will add these clarifications to the revision.
>
> **Q2: Computational overhead.**
>
> Thanks for this insightful comment!
>
> We measure the prefill stage latency on an NVIDIA GeForce RTX 4090 over 500 samples, as the computational overhead of SinkTrack is **strictly confined** to this stage. The results are shown below:
>
> | **Model (Latency)** | **w/o Injection** | **w/ Injection** | **Overhead** |
> | --- | --- | --- | --- |
> | Llama3.1-8B | 35.90 ms | 36.66 ms | +0.76 ms |
> | Qwen2.5-VL-7B | 107.13 ms | 110.26 ms | +3.13 ms |
>
> These findings confirm that SinkTrack introduces **negligible overhead**, as stated in our paper (L84-86):
>
> > Its core intervention is a one-off, lightweight operation performed prior to generation. Since the intervention can be stored in KV cache, SinkTrack adds negligible computational overhead to the subsequent auto-regressive steps, thereby preserving LLMs' original inference speed.
> >
>
> We will include this analysis in the revision.
>
> **Q3:  CoT performance.**
>
> Thanks for your feedback. The observation that CoT can underperform Direct Prompting is a well-documented phenomenon in the field.
>
> For example, recent studies find that for long-context understanding tasks like document summarization and QA, CoT can even degrade performance [ref3-4]. Additionally, while CoT excels in complex reasoning [ref1], as shown in our M3CoT results, for non-reasoning tasks, generating long chains is counterproductive. It exacerbates long-context issues like attention drift and raises the risk of error accumulation, where a single misstep derails the final answer [ref2-3].
>
> We add a concise discussion to the revision to explain these results:
>
> > We observe that CoT sometimes underperforms Direct Prompting, an observation consistent with existing studies. This performance degradation is highly task-dependent. In long-context understanding tasks, such as summarization and QA, CoT can even degrade performance [ref3-4]. Furthermore, while CoT excels in complex reasoning [ref1] like M3CoT, in non-reasoning tasks, its extended chains exacerbate attention drift and raise the risk of error accumulation, where a single misstep derails the final answer [ref2-3].
> >

---

> ### Author Response · Authors · 2025-11-20
> **Response to Reviewer 85VA (2/3)**
>
> **Q4：Limited baselines.**
>
> Good suggestion!
>
> We now include **comparisons with recent, open-source SOTA training-free methods in both modalities**. For multi-modal tasks, we benchmark against **VCD [ref5], OPERA [ref11], ICD [ref6], and SID [ref7]** on the POPE:
>
> - **VCD** mitigates reliance on language priors by contrasting output distributions from original versus distorted images.
> - **OPERA** counters the "over-trust" issue by applying a logit penalty and retrospection-allocation strategy during decoding, reducing over-reliance on summary tokens over image content.
> - **ICD** amplifies visual information by contrasting outputs generated from original versus disturbance-added instructions.
> - **SID** employs a self-introspection mechanism, leveraging the contrast between an initial draft and a self-critiqued revision to guide factual generation.
>
> | **Method** | **Accuracy (%)** | **F1-score** |
> | --- | --- | --- |
> | VCD [ref5] | 84.63 | 82.83 |
> | OPERA [ref11] | 84.67 | 83.01 |
> | ICD [ref6] | 84.71 | 82.94 |
> | SID [ref7] | 85.24 | 85.36 |
> | **SinkTrack (Ours)** | **86.52** | **86.05** |
>
> For text-modal tasks, we conduct **new experiments on LongBench** [ref9]. LongBench comprises 6 major task categories and 20 sub-tasks, with inputs **ranging from 5k to 15k** tokens in length. We compare against **InfLLM [ref8]** on LongBench. InfLLM stores distant contexts into additional memory units and employs an efficient mechanism to lookup token-relevant units for attention computation.
>
> | **Model (Llama3.1-8B)** | **0-4k** | **4-8k** | **8k+** | **Average** |
> | --- | --- | --- | --- | --- |
> | Direct | 39.58 | 40.54 | 43.11 | 41.08 |
> | CoT | 35.91 | 38.35 | 41.03 | 38.43 |
> | InfLLM [ref8] | 36.73 | 37.19 | 39.77 | 39.23 |
> | **SinkTrack (Ours)** | **40.43** | **41.66** | **44.32** | **42.14** |
>
> | **Model (Qwen2.5-7B)** | 0-4k | 4-8k | 8k+ | total |
> | --- | --- | --- | --- | --- |
> | Direct | 40.17 | 44.32 | 46.92 | 43.8 |
> | CoT | 33.42 | 38.05 | 40.03 | 37.17 |
> | InfLLM [ref8] | 35.7 | 38.62 | 38.84 | 37.72 |
> | **SinkTrack (Ours)** | **40.58** | **44.9** | **47.52** | **44.34** |
>
> We will include the new results in the revision.
>
> It is worth noting that InfLLM is originally developed for Llama2 and Qwen1.5. To ensure a fair comparison on newer architectures, we implement it as follows: For Llama3.1, we utilized the subsequent code extension **released by the authors**. For Qwen2.5, we analyze the source code and confirmed that the interface compatibility allows for direct adaptation (e.g., updating the class invocation from QwenModel to Qwen2Model). Despite this valid adaptation, its performance remains suboptimal, as confirmed by **community feedback [ref10]** and our results.

---

> ### Author Response · Authors · 2025-11-20
> **Response to Reviewer 85VA (3/3)**
>
> **Q5: Ambiguity of "vertical" and "horizontal" information flow**
>
> Our apologies.
>
> We now provide more detailed definitions:
>
> - **Vertical Information Flow** measures how the injected information propagates **through the Transformer layers** during the input processing (prefill) stage. We measure the "Information Gain" for each token at each layer by computing the L2 norm between its hidden state vectors *with* and *without* injection. The L2 norm quantifies the magnitude of change in the vector space, answering "**by how much was the token shifted?**" A larger distance signifies a stronger impact and more effective propagation.
> - **Horizontal Information Flow** measures how anchored information influences **newly generated tokens**. To quantify this influence, we compute the L2 norm of the difference between the hidden vectors (prior to vocabulary projection) of each newly generated token under the two modes (*with* vs. *without* injection). To ensure a fair comparison, the analysis is always conducted on **identical** token sequences. If the first generated token differs between the two modes, we will force the subsequent generation to follow the path of the "*with* injection" mode.
>
> These more detailed definitions are added to **Section 3.4.3** of the revision.
>
> **Q6: Potential information loss and generalizability.**
>
> Thanks for your feedback. We acknowledge that compressing context into a single token presents an information bottleneck.
>
> However, surprisingly, our **extensive new experiments** demonstrate SinkTrack’s practical utility. Our evaluation is **NOT** limited to **two tasks** but spans **a wide array of challenges** across vision and text modalities. This includes complex multi-hop reasoning (M3CoT), general capabilities (MMStar, with 18 sub-tasks), real-world VQA (RealWorldQA), hallucination detection (POPE), dialogue and context understanding (QuAC, SQuAD v2).
>
> Furthermore, we now add a diverse suite of tasks in **LongBench** (covering 6 major categories and 20 sub-tasks, including **single/multi-document QA, summarization, few-shot learning, and code completion, etc**.). The results of LongBench are shown as Q4.
>
> Further modeling and addressing this potential bottleneck will be a top priority in our future work.
>
> [ref1] To CoT or not to CoT? Chain-of-thought helps mainly on math and symbolic reasoning, ICLR 2025
>
> [ref2] Adaptive Chain-of-Thought Distillation Based on LLM Performance on Original Problems, Mathematics 2025
>
> [ref3] Mind Your Step (by Step): Chain-of-Thought can Reduce Performance on Tasks where Thinking Makes Humans Worse, arxiv 2024
>
> [ref4] Unable to Forget: Proactive Interference Reveals Working Memory Limits in LLMs Beyond Context Length, ICML 2025
>
> [ref5] VCD: Mitigating Object Hallucinations in Large Vision-Language Models through Visual Contrastive Decoding, CVPR 2024
>
> [ref6] Mitigating Hallucinations in Large Vision-Language Models with Instruction Contrastive Decoding, ACL 2024
>
> [ref7] Self-Introspective Decoding: Alleviating Hallucinations for Large Vision-Language Models, ICLR 2025
>
> [ref8] InfLLM: Training-Free Long-Context Extrapolation for LLMs with an Efficient Context Memory, NeurIPS 2024
>
> [ref9] LongBench: A Bilingual, Multitask Benchmark for Long Context Understanding, ACL 2024
>
> [ref10] https://github.com/thunlp/InfLLM/issues/51
>
> [ref11] OPERA: Alleviating Hallucination in Multi-Modal Large Language Models via Over-Trust Penalty and Retrospection-Allocation, CVPR 2024

---

> > ### Comment · Reviewer_85VA · 2025-11-24
> > **Response to Author's Rebuttal**
> >
> > Thanks for the response. Currently I have no more questions. However, as the reproduction issue has been raised by other researchers. I choose to maintain my current score until it has been solved.

---

> > > ### Author Response · Authors · 2025-11-25
> > >
> > > Thanks again for your constructive comments! To ensure full reproducibility, we provide complete artifacts in our anonymous repository. We upload all code, inference results, and evaluation scripts so that every reported number can be recomputed from our official implementation. Pls see README for step-by-step instructions. If you encounter any issues or have any further concerns, we would be very happy to clarify details or provide additional materials.

---

> > > > ### Comment · Reviewer_85VA · 2025-11-28
> > > > **Response to Authors**
> > > >
> > > > Thanks for the reply. I have checked all relevant messages between the authors and other reviewers and believe there is no significant reproducibility problem. I have decided to maintain my current overall score, as the paper requires revisions in both the methodology and experimental sections to resolve confusion and strengthen the empirical evidence. However, I am reducing my confidence score from 4 to 3.

---

> > > > > ### Author Response · Authors · 2025-11-28
> > > > > **Response to Reviewer 85VA (2/3)**
> > > > >
> > > > > **PART 3. Validation on New Model**
> > > > >
> > > > > As shown in PART 2, we extend our evaluation to **LLaVA-v1.5-7B** to demonstrate generalizability across different multi-modal architectures.
> > > > >
> > > > > **PART 4. Robustness Verification**
> > > > >
> > > > > Finally, to demonstrate the robustness of SinkTrack, we re-run experiments across datasets and models using **three different random seeds (323, 500, 900)** and now report results as **mean ± variance**.
> > > > >
> > > > > |  |  | Qwen2.5-VL-3B |  | Gemma3-4B |  | Qwen2.5-VL-7B |  | Gemma3-12B |  |
> > > > > | --- | --- | --- | --- | --- | --- | --- | --- | --- | --- |
> > > > > |  |  | Acc | F1 | Acc | F1 | Acc | F1 | Acc | F1 |
> > > > > | M3CoT | **Direct** | 33.43±0.54 | 44.2±3.08 | 27.71±0.20 | 33.81±0.62 | 39.2±0.29 | 51.01±0.55 | 41.9±0.40 | 51.46±0.87 |
> > > > > |  | **Cot** | 20.69±0.51 | 32.04±0.62 | 50.96±0.32 | **59.71±1.03** | 44.11±0.56 | 57.96±0.96 | 60.32±0.30 | **72.24±0.83** |
> > > > > |  | **SinkTrack** | **48.25±0.09** | **58.73±1.14** | **51.01±0.78** | 59.18±1.2 | **66.94±0.21** | **79.13±0.23** | **60.57±0.12** | 70.97±0.67 |
> > > > > |  |  |  |  |  |  |  |  |  |  |
> > > > > | MMstar | **Direct** | 24.89±1.65 | 39.54±1.90 | 35.47±0.20 | 52.17±0.12 | 37.29±0.80 | 54.24±0.95 | 44.58±0.55 | 60.89±0.59 |
> > > > > |  | **Cot** | 35.24±0.83 | 50.68±0.95 | 51.36±1.53 | 67.52±1.32 | 55.84±0.74 | 71.29±0.73 | 60.56±0.91 | 75.09±0.67 |
> > > > > |  | **SinkTrack** | **47.24±0.40** | **61.38±0.44** | **53.04±0.53** | **69.11±0.48** | **63.78±0.19** | **77.86±0.17** | **61.6±0.46** | **75.81±0.51** |
> > > > > |  |  |  |  |  |  |  |  |  |  |
> > > > > | POPE | **Direct** | 53.23±0.24 | 69.48±0.20 | 83.98±0.04 | 91.3±0.02 | 78.21±0.27 | 87.15±0.20 | 84.59±0.07 | 91.65±0.04 |
> > > > > |  | **Cot** | 64.58±0.22 | 78.1±0.16 | 82.01±0.38 | 90.04±0.24 | 83.65±0.08 | 90.63±0.06 | 84.53±0.33 | 91.61±0.19 |
> > > > > |  | **SinkTrack** | **77.69±0.12** | **87.44±0.07** | **84.33±0.02** | **91.5±0.01** | **85.47±0.02** | **91.67±0.01** | **85.4±0.13** | **92.13±0.05** |
> > > > > |  |  |  |  |  |  |  |  |  |  |
> > > > > | RealWorldQA | **Direct** | 31.15±1.63 | 23.25±4.36 | 51.33±0.06 | **48.43±1.10** | 47.49±0.22 | 39.93±4.33 | 57.34±0.44 | 50.96±3.74 |
> > > > > |  | **Cot** | 35.69±1.61 | 26.46±4.71 | 49.41±1.02 | 44.53±5.27 | 57.86±1.29 | 43±4.14 | 57.69±0.25 | 49.92±2.00 |
> > > > > |  | **SinkTrack** | **48.28±0.40** | **28.42±0.22** | **52.46±0.43** | 47.39±1.59 | **65.49±0.37** | **52.81±0.68** | **57.91±0.33** | **52.54±0.77** |
> > > > >
> > > > > |  |  | MiniCPM3-4B |  | Qwen2.5-7B |  | Llama3.1-8B |  |
> > > > > | --- | --- | --- | --- | --- | --- | --- | --- |
> > > > > |  |  | **Acc** | **F1** | **Acc** | **F1** | **Acc** | **F1** |
> > > > > | QuAC-1 | **Direct** | 52.82±0.62 | 52.79±0.4 | 52.57±0.39 | 47.06±0.2 | 54.1±0.02 | 48.87±0.14 |
> > > > > |  | **CoT** | 42.69±0.94 | 45.45±0.71 | 38.47±1.55 | 42.11±0.9 | 54.9±0.12 | 46.09±0.09 |
> > > > > |  | **SinkTrack** | **53.57±0.41** | **52.82±0.43** | **52.93±0.25** | **47.52±0.21** | **59.4±0.1** | **54.05±0.04** |
> > > > > |  |  |  |  |  |  |  |  |
> > > > > | QuAC-2 | **Direct** | 49.97±0.34 | 50.94±0.23 | 52±0.2 | 48.06±0.1 | 54.25±0.08 | 51.31±0.09 |
> > > > > |  | **CoT** | 41.03±0.33 | 43.35±0.73 | 37.3±1.16 | 39.42±0.76 | 51.75±0.24 | 38.21±0.11 |
> > > > > |  | **SinkTrack** | **50.12±0.26** | **50.98±0.25** | **52.98±0.1** | **48.08±0.07** | **58.05±0.05** | **54.25±0.08** |
> > > > > |  |  |  |  |  |  |  |  |
> > > > > | QuAC-3 | **Direct** | 48.13±0.36 | 49.53±0.23 | 51.26±0.22 | 47.64±0.15 | 53.7±0.04 | 51.2±0.13 |
> > > > > |  | **CoT** | 39.39±0.27 | 41.19±0.74 | 36.34±0.68 | 37.92±0.22 | 49.13±0.27 | 34.47±0.13 |
> > > > > |  | **SinkTrack** | **48.53±0.15** | **49.75±0.21** | **52.01±0.3** | **47.77±0.11** | **56.2±0.09** | **53.45±0.11** |
> > > > > |  |  |  |  |  |  |  |  |
> > > > > | QuAC | **Direct** | 47.17±0.19 | 48.07±0.05 | 49.47±0.07 | 46.71±0.15 | 52.45±0.07 | 50.66±0.15 |
> > > > > |  | **CoT** | 37.68±0.18 | 36.5±0.2 | 36.48±0.29 | 36.58±0.09 | 46.95±0.29 | 28.74±0.05 |
> > > > > |  | **SinkTrack** | **47.29±0.27** | **48.08±0.3** | **50.25±0.21** | **46.73±0.15** | **53.51±0.06** | **51.56±0.09** |

---

> > > > > ### Author Response · Authors · 2025-11-28
> > > > > **Response to Reviewer 85VA (3/3)**
> > > > >
> > > > > **PART 5. Additional Analytical Experiments**
> > > > >
> > > > > To resolve methodological ambiguity and verify efficiency, we conduct further specific analyses:
> > > > >
> > > > > - **SinkTrack Drift Test:** We analyze the attention patterns of Llama3.1-8B by feeding it an **input context of 8k+ tokens** and tracking the attention weights allocated to <BOS> by **subsequently generated tokens** (from step 1 to 1024). The results show that the <BOS> weight remains dominant (exceeding the sum of all other tokens), proving that SinkTrack is **immune to attention drift**.
> > > > >
> > > > >
> > > > >     | **Generated Token Index ($i$)** | **Actual Sequence Position ($L_{input} + i$)** | **Attention from Token $i$ to <BOS>** | **Max Attention from Token $i$ to Others** | **Ratio (BOS / Max)** |
> > > > >     | --- | --- | --- | --- | --- |
> > > > >     | 1 | 8083 | 0.561 | 0.057 | 9.87x |
> > > > >     | 64 | 8146 | 0.570 | 0.048 | 11.97x |
> > > > >     | 128 | 5210 | 0.612 | 0.054 | 11.25x |
> > > > >     | 256 | 8338 | 0.560 | 0.050 | 11.24x |
> > > > >     | 512 | 8594 | 0.558 | 0.037 | 15.17x |
> > > > >     | 1024 | 9106 | 0.582 | 0.042 | 13.96x |
> > > > > - **Overhead Measurement:** We measure the prefill stage latency on an NVIDIA GeForce RTX 4090 over 500 samples:
> > > > >
> > > > >
> > > > >     | **Model (Latency)** | **w/o Injection** | **w/ Injection** | **Overhead** |
> > > > >     | --- | --- | --- | --- |
> > > > >     | Llama3.1-8B | 35.90 ms | 36.66 ms | +0.76 ms |
> > > > >     | Qwen2.5-VL-7B | 107.13 ms | 110.26 ms | +3.13 ms |
> > > > >
> > > > >     These findings confirm that SinkTrack introduces **negligible overhead**, as stated in our paper (L84-86):
> > > > >
> > > > >     > Its core intervention is a one-off, lightweight operation performed prior to generation. Since the intervention can be stored in KV cache, SinkTrack adds negligible computational overhead to the subsequent auto-regressive steps, thereby preserving LLMs' original inference speed.
> > > > >     >
> > > > >
> > > > > We will include all of these results and analysis in our revision.
> > > > >
> > > > > We comprehensively update the **Methodology** and **Experiment** sections in our revision to reflect these improvements and resolve any prior confusion. We wish these revisions strengthen the paper, and we warmly welcome your any additional suggestions to further improve our manuscript.
> > > > >
> > > > > [ref1] LongBench: A Bilingual, Multitask Benchmark for Long Context Understanding, ACL 2024
> > > > >
> > > > > [ref2] InfLLM: Training-Free Long-Context Extrapolation for LLMs with an Efficient Context Memory, NeurIPS 2024
> > > > >
> > > > > [ref3] Learn to Explain: Multimodal Reasoning via Thought Chains for Science Question Answering, NeurIPS 2022
> > > > >
> > > > > [ref4] Scaffolding Coordinates to Promote Vision-Language Coordination in Large Multi-Modal Models, COLING 2025
> > > > >
> > > > > [ref5] LLM Maybe LongLM: Self-Extend LLM Context Window Without Tuning, ICML 2024
> > > > >
> > > > > [ref6] VCD: Mitigating Object Hallucinations in Large Vision-Language Models through Visual Contrastive Decoding, CVPR 2024
> > > > >
> > > > > [ref7] OPERA: Alleviating Hallucination in Multi-Modal Large Language Models via Over-Trust Penalty and Retrospection-Allocation, CVPR 2024
> > > > >
> > > > > [ref8] Mitigating Hallucinations in Large Vision-Language Models with Instruction Contrastive Decoding, ACL 2024
> > > > >
> > > > > [ref9] Self-Introspective Decoding: Alleviating Hallucinations for Large Vision-Language Models, ICLR 2025

---

> ### Author Response · Authors · 2025-11-28
> **Response to Reviewer 85VA (1/3)**
>
> Thanks for your response and your time. We are encouraged to hear that your concerns regarding reproducibility have been resolved.
>
> We agree that better evidence and clearer methodology are crucial. To this end, we incorporate a substantial amount of new results into the revision, covering **new benchmarks, competitive baselines, additional model architectures, robust stability tests, and additional analytical experiments.**.
>
> **PART 1. Expansion of Benchmarks**
>
> Following suggestions from reviewers, we expand our evaluation to include **LongBench [ref1]** (for comprehensive long-context understanding) and **ScienceQA [ref3]** (for multi-modal reasoning).
>
> | **LongBench\\Llama3.1-8B** | **0-4k** | **4-8k** | **8k+** | **total** |
> | --- | --- | --- | --- | --- |
> | Direct | 39.58 | 40.54 | 43.11 | 41.08 |
> | CoT | 35.91 | 38.35 | 41.03 | 38.43 |
> | InfLLM [ref2] | 36.73 | 37.19 | 39.77 | 39.23 |
> | LongLM [ref5] | 38.74 | 39.01 | 41.53 | 39.76 |
> | **SinkTrack** | **40.43** | **41.66** | **44.32** | **42.14** |
>
> |**LongBench\\Qwen2.5-7B** | **0-4k** | **4-8k** | **8k+** | **total** |
> | --- | --- | --- | --- | --- |
> | Direct | 39.58 | 40.54 | 43.11 | 41.08 |
> | CoT | 35.91 | 38.35 | 41.03 | 38.43 |
> | InfLLM [ref2] | 36.73 | 37.19 | 39.77 | 39.23 |
> | LongLM [ref5] | 39.88 | 42.78 | 45.84 | 42.83 |
> | **SinkTrack** | **40.58** | **44.90** | **47.52** | **44.34** |
>
> | **ScienceQA\\Qwen2.5-VL-7B** | **Acc** | **F1** |
> | --- | --- | --- |
> | Direct | 87.56 | 82.68 |
> | CoT | 89.34 | 88.77 |
> | Scaffold [ref4] | 85.52 | **90.44** |
> | **SinkTrack** | **90.53** | 89.34 |
>
> SinkTrack consistently outperforms baselines.
>
> **PART 2. Comparison with Stronger Baselines**
>
> We also compare SinkTrack against a wider range of competitive, training-free SOTA baselines:
>
> - **For Long Context:** **InfLLM [ref2]**, **LongLM** **[ref5]** on LongBench [ref1] (as shown in PART 1), QuAC and SQuAD2.0 (as shown below):
>
>
>     |  |  | Acc | F1 |
>     | --- | --- | --- | --- |
>     | **QuAC-1** | **Direct** | 54.1±0.02 | 48.87±0.14 |
>     |  | **CoT** | 54.9±0.12 | 46.09±0.09 |
>     |  | **LongLM** [ref5] | 53.8 | 48.82 |
>     |  | **SinkTrack** | **59.4±0.1** | **54.05±0.04** |
>     |  |  |  |  |
>     | **QuAC-2** | **Direct** | 54.25±0.08 | 51.31±0.09 |
>     |  | **CoT** | 51.75±0.24 | 38.21±0.11 |
>     |  | **LongLM** [ref5] | 54.1 | 51.2 |
>     |  | **SinkTrack** | **58.05±0.05** | **54.25±0.08** |
>     |  |  |  |  |
>     | **QuAC-3** | **Direct** | 53.7±0.04 | 51.2±0.13 |
>     |  | **CoT** | 49.13±0.27 | 34.47±0.13 |
>     |  | **LongLM** [ref5] | 53.4 | 51.09 |
>     |  | **SinkTrack** | **56.2±0.09** | **53.45±0.11** |
>     |  |  |  |  |
>     | **QuAC** | **Direct** | 52.45±0.07 | 50.66±0.15 |
>     |  | **CoT** | 46.95±0.29 | 28.74±0.05 |
>     |  | **LongLM** [ref5] | 52.52 | 50.81 |
>     |  | **SinkTrack** | **53.51±0.06** | **51.56±0.09** |
>     | **SQuAD2.0** | **Direct** | 78.69±0.02 | 24.95±0.04 |
>     |  | **CoT** | 58.19±0.31 | 28.86±0.07 |
>     |  | **LongLM** [ref5] | 78.77 | 24.9 |
>     |  | **SinkTrack** | **79.83±0.07** | **25.72±0.13** |
> - **For Multi-modal Hallucination:** **VCD** [ref6], **OPERA** [ref7]**, ICD** [ref8], **SID** [ref9] (as shown below) and **Scaffold [ref4]** (as shown in PART 1):
>
>
>     | **POPE/LLaVA-v1.5-7B** | **Accuracy (%)** | **F1-score** |
>     | --- | --- | --- |
>     | VCD [ref1] | 84.63 | 82.83 |
>     | OPERA [ref7] | 84.67 | 83.01 |
>     | ICD [ref2] | 84.71 | 82.94 |
>     | SID [ref3] | 85.24 | 85.36 |
>     | **SinkTrack (Ours)** | **86.52** | **86.05** |

---

### Official Review · Reviewer_Utsb · 2025-10-19

**Soundness:** 2
**Presentation:** 2
**Contribution:** 3
**Rating:** 4
**Confidence:** 2

**Summary:**

This paper tackles hallucination and context forgetting caused by attention drift. The authors observe that, during generation, LLMs gradually shift their focus toward newly generated tokens and away from the initial context. Inspired by attention sink, they propose SinkTrack, a training-free, inference-time enhancement that treats the BOS token as an information anchor and continually injects key contextual features (e.g., image or instruction embeddings) into its representation. Experiments on both text-only and multi-modal QA show that SinkTrack suppresses drift-induced hallucinations and forgetting, outperforming both direct inference and chain-of-thought reasoning.

**Strengths:**

1. The paper highlights how attention drift triggers hallucination and context forgetting, which is a key insight for both long-context LLMs and long-context multi-modal LLMs.
2. The authors introduce a novel, training-free inference enhancement that alleviates the above issues with negligible overhead.
3. The effectiveness is demonstrated across the Qwen, Gemma, MiniCPM, and Llama families, accompanied by visualization analyses and discussions.

**Weaknesses:**

1. The evaluation benchmarks are somewhat outdated. While my limited knowledge of vision-related tasks and hallucination-related discussion prevents me from assessing those aspects, relying solely on QuAC and SQuAD to evaluate SinkTrack’s long-context improvements is clearly insufficient. The authors should first justify why QuAC is representative and then add results on standard long-context benchmarks such as LongBench [1], L-Eval [2], NIAH [3], and RULER [4], or more recent long-term conversation benchmarks like LongMemEval [5].
2. The authors compare SinkTrack only against direct query and Chain-of-Thought, without comparison with the methods cited in Related Work. While three main streams of prior work are reviewed, their empirical relationship to SinkTrack has not been examined, such as an experimental comparison with a retrieval-augmented method. Including such comparisons would greatly strengthen the cohesion of the paper and the soundness of the method.
3. The evaluation does not state the context lengths used, nor does it examine how attention drift varies as prompt length grows. Because SinkTrack itself relies on attention-based calculation, I am worried about whether the attention for SinkTrack could also drift when the prompt becomes very long.

[1] LongBench: A Bilingual, Multitask Benchmark for Long Context Understanding https://arxiv.org/abs/2308.14508

[2] L-Eval: Instituting Standardized Evaluation for Long Context Language Models https://arxiv.org/abs/2307.11088

[3] Needle In A Haystack - Pressure Testing LLMs https://github.com/gkamradt/LLMTest_NeedleInAHaystack

[4] RULER: What's the Real Context Size of Your Long-Context Language Models? https://arxiv.org/abs/2404.06654

[5] LongMemEval: Benchmarking Chat Assistants on Long-Term Interactive Memory https://arxiv.org/abs/2410.10813

**Questions:**

See Weaknesses

---

> ### Author Response · Authors · 2025-11-20
> **Response to Reviewer Utsb (1/2)**
>
> We thank reviewer Utsb for the valuable time and constructive feedback. We provide point-to-point response below.
>
> **Q1: Outdated evaluation benchmarks.**
>
> Good suggestion!
>
> Following your advice, we conduct a **evaluation on LongBench [ref1] using Llama3.1-8B and Qwen2.5-7B.**
>
> We also include a comparison with the **SOTA method InfLLM [ref2]**. InfLLM stores distant contexts into additional memory units and employs an efficient mechanism to lookup token-relevant units for attention computation.
>
> | **Model (Llama3.1-8B)** | **0-4k** | **4-8k** | **8k+** | **Average** |
> | --- | --- | --- | --- | --- |
> | Direct | 39.58 | 40.54 | 43.11 | 41.08 |
> | CoT | 35.91 | 38.35 | 41.03 | 38.43 |
> | InfLLM **[ref2]** | 36.73 | 37.19 | 39.77 | 39.23 |
> | **SinkTrack (Ours)** | **40.43** | **41.66** | **44.32** | **42.14** |
>
> | **Model (Qwen2.5-7B)** | 0-4k | 4-8k | 8k+ | **Average** |
> | --- | --- | --- | --- | --- |
> | Direct | 40.17 | 44.32 | 46.92 | 43.8 |
> | CoT | 33.42 | 38.05 | 40.03 | 37.17 |
> | InfLLM **[ref2]** | 35.7 | 38.62 | 38.84 | 37.72 |
> | **SinkTrack (Ours)** | **40.58** | **44.9** | **47.52** | **44.34** |
>
> We will include the new results in the revision.
>
> It is worth noting that InfLLM is originally developed for Llama2 and Qwen1.5. To ensure a fair comparison on newer architectures, we implement it as follows: For Llama3.1, we utilized the subsequent code extension **released by the authors**. For Qwen2.5, we analyze the source code and confirmed that the interface compatibility allows for direct adaptation (e.g., updating the class invocation from QwenModel to Qwen2Model). Despite this valid adaptation, its performance remains suboptimal, as confirmed by **community feedback [ref3]** and our results.
>
> Regarding the performance of the CoT baseline, please refer to our detailed analysis in the responses to **Reviewer 4 (85VA) Q3** and **Reviewer 5 (YQZu) Q7**.
>
> **Q2: QuAC’s representation.**
>
> Thanks for bringing this up for discussion.
>
> We choose QuAC because:
>
> 1. It is a widely adopted benchmark in official evaluations of mainstream LLMs (eg., Llama 3.1).
> 2. Its context grows longer with each dialogue turn. This structure is ideal for evaluating context forgetting, as stated in our paper (L264-266):
>
>     > Its structure, where questions build upon the dialogue history, allows us to analyze performance across progressive dialogue turns (QuAC-1, -2, -3) as the context lengthens, thereby assessing the mitigation of context forgetting.
>     >
>
> We will articulate this justification more clearly in the revision.

---

> ### Author Response · Authors · 2025-11-20
> **Response to Reviewer Utsb (2/2)**
>
> **Q3: Lack of baselines.**
>
> Thanks for this insightful comment!
>
> We now include **comparisons with recent, open-source SOTA training-free methods in both modalities**. For multi-modal tasks, we benchmark against **VCD [ref4], OPERA [ref9], ICD [ref5], and SID [ref6]** on the POPE:
>
> - **VCD** mitigates reliance on language priors by contrasting output distributions from original versus distorted images.
> - **OPERA** mitigates the "over-trust" issue by introducing a logit penalty and a retrospection-allocation strategy during decoding, countering the model's tendency to excessively focus on summary tokens rather than image content.
> - **ICD** amplifies visual information by contrasting outputs generated from original versus disturbance-added instructions.
> - **SID** employs a self-introspection mechanism, leveraging the contrast between an initial draft and a self-critiqued revision to guide factual generation.
>
> For text-modal tasks, we compare against **InfLLM** on LongBench (as shown in Q1). In all comparisons, SinkTrack demonstrates superior performance.
>
> | **Method** | **Accuracy (%)** | **F1-score** |
> | --- | --- | --- |
> | VCD [ref4] | 84.63 | 82.83 |
> | OPERA [ref9] | 84.67 | 83.01 |
> | ICD [ref5] | 84.71 | 82.94 |
> | SID [ref6] | 85.24 | 85.36 |
> | **SinkTrack (Ours)** | **86.52** | **86.05** |
>
> We will include the new results in the revision.
>
> **Q4: Potential for SinkTrack to drift.**
>
> Good question!
>
> SinkTrack's stability is grounded in the "attention sink" phenomenon. Prior research [ref7-8] establishes this as an **intrinsic property of LLM Transformer architectures**, persisting consistently across **diverse model families, modalities, and sequence lengths**.
>
> To empirically address your concern, we analyze the attention patterns of Llama3.1-8B. We feed the model an **input context of 8k+ tokens** and then track the attention weights allocated to \<BOS\> by the **subsequently generated tokens** (from step 1 to 1024).
>
> | **Generated Token Index ($i$)** | **Actual Sequence Position ($L_{input} + i$)** | **Attention from Token $i$ to \<BOS\>** | **Max Attention from Token $i$ to Others** | **Ratio (BOS / Max)** |
> | --- | --- | --- | --- | --- |
> | 1 | 8083 | 0.561 | 0.057 | 9.87x |
> | 64 | 8146 | 0.570 | 0.048 | 11.97x |
> | 128 | 5210 | 0.612 | 0.054 | 11.25x |
> | 256 | 8338 | 0.560 | 0.050 | 11.24x |
> | 512 | 8594 | 0.558 | 0.037 | 15.17x |
> | 1024 | 9106 | 0.582 | 0.042 | 13.96x |
>
> **It directly alleviates drift concerns.** As shown above, even after 8k+ tokens, the attention to \<BOS\> remains exceptionally high—**exceeding** both **10x** the maximum attention to any other token and the **sum** of all other tokens' attention. This confirms that SinkTrack maintains a robust, non-degrading anchor point in long contexts.
>
> [ref1] LongBench: A Bilingual, Multitask Benchmark for Long Context Understanding, ACL 2024
>
> [ref2] InfLLM: Training-Free Long-Context Extrapolation for LLMs with an Efficient Context Memory, NeurIPS 2024
>
> [ref3] https://github.com/thunlp/InfLLM/issues/51
>
> [ref4] VCD: Mitigating Object Hallucinations in Large Vision-Language Models through Visual Contrastive Decoding, CVPR 2024
>
> [ref5] Mitigating Hallucinations in Large Vision-Language Models with Instruction Contrastive Decoding, ACL 2024
>
> [ref6] Self-Introspective Decoding: Alleviating Hallucinations for Large Vision-Language Models, ICLR 2025
>
> [ref7] “Why do LLMs attend to the first token?”, COLM 2025
>
> [ref8] “Efficient Streaming Language Models with Attention Sinks”, ICLR 2024
>
> [ref9] OPERA: Alleviating Hallucination in Multi-Modal Large Language Models via Over-Trust Penalty and Retrospection-Allocation, CVPR 2024

---

> > ### Comment · Reviewer_Utsb · 2025-11-25
> >
> > Thank you for the patient reply. Evaluations based on more widely used benchmarks and newer checkpoints are more convincing in demonstrating the effectiveness of your method. We appreciate your efforts in adapting some methods to new models, although I somehow regard InfLLM as more of an efficient method than an inference enhancement method.
> >
> > The new experiments on different generation lengths validate the hypothesis of SinkTrack, which I think is crucial for enhancing the robustness of your method. Thank you for the additional information. Despite my limited knowledge of vision-related tasks and hallucination-related discussion, I raise my rating to 6 to acknowledge your sincere response.

---

> > > ### Author Response · Authors · 2025-11-25
> > >
> > > Thank you for your valuable time and constructive comments. We are deeply grateful for your recognition of our rebuttal efforts.
> > >
> > > Your suggestions regarding the adoption of standard long-context benchmarks (e.g., LongBench) and the examination of attention drift across varying generation lengths are instrumental. Incorporating these experiments significantly strengthen the empirical persuasiveness of SinkTrack and provide validation for the robustness of it. All these analyses and results inspired by your suggestions will be fully integrated into the revision.
> > >
> > > We remain fully open to any further discussion during the rebuttal period should you have additional questions. Thanks.

---

> ### Author Response · Authors · 2025-11-25
>
> Dear reviewer Utsb,
>
> **To ensure full reproducibility, we provide complete artifacts in our [anonymous repository](https://anonymous.4open.science/r/iclr2026sinktrack-20040323/README.md).** We upload all **code**, **inference results**, and **evaluation scripts** so that every reported number can be recomputed from our official implementation. Pls see README for step-by-step instructions. If you encounter any issues or have any further concerns, we would be very happy to clarify details or provide additional materials.

---

### Official Review · Reviewer_DYoN · 2025-10-31

**Soundness:** 3
**Presentation:** 3
**Contribution:** 2
**Rating:** 4
**Confidence:** 3

**Summary:**

This paper proposes a training-free method to enable context anchoring (reducing hallucination) for language models, the key idea is to leverage the attention sink phenomena and fuse the hidden representation of context into the sink token to encourage more attention to the context.

Experiments are conducted on three LLMs and four VLMs, showing improved performance over baseline (direct inference and CoT inference).

**Strengths:**

* The paper is written relatively clearly.
* The idea of leveraging attention sink to encourage context anchoring is interesting.

**Weaknesses:**

* The baseline is limited: Using attention sink to enhance context anchoring is nice, yet this paper does not compare to prior training-free methods to mitigate halluciation. To name a few: [ACT(ICML 2024)](https://arxiv.org/pdf/2406.15765), [VCD(CVPR 2024)](https://arxiv.org/pdf/2311.16922), [OPERA(CVPR 2024)](https://arxiv.org/pdf/2311.17911), [SID(ICLR 2025)](https://arxiv.org/pdf/2408.02032), [DAC(ICML 2025)](https://arxiv.org/abs/2502.01969).

* It is nice that the paper evaluates on both image and textual benchmarks, yet i found the choice of textual benchmark a bit unconventional. If the idea is to mitigate hallucination for long context, evaluation on well-studied benchmarks such as [LongBench](https://arxiv.org/abs/2308.14508) would be appropriate.

**Questions:**

* IIUC, the SINKTRACK method lets the <BOS> token attend to the rest of the tokens (instead of replacing the K,V of its own with the mean pooling of the rest of the token). If so, that should be illustrated clearly in Figure 3 (which shows the mean pooling operation).
* I would suggest the author to present their full methods more clearly. Currently the paper is written to illustrate the "failed" attempt first (hard injection, soft injection, mean pooling), which makes it a bit hard to understand the actual proposed method. These comparison can be included in the ablation study.

---

> ### Author Response · Authors · 2025-11-20
> **Response to Reviewer DYoN (1/2)**
>
> We thank reviewer DYoN for the valuable time and constructive feedback. We provide point-to-point response below.
>
> **Q1: Limited baselines.**
>
> Thanks for your suggestion.
>
> Following your suggestion, we benchmark SinkTrack against the **four publicly available SOTA training-free methods (VCD [ref1], OPERA [ref7], ICD [ref2], and SID [ref3])** on the POPE using LLaVA-v1.5-7B.
>
> We select these baselines based on code availability and compatibility to ensure a fair comparison. Specifically, we exclude DAC as its code is not open-source; ACT because its official implementation is limited to outdated models (e.g., Llama 2).
> In addition, we include **ICD [ref2]** as a supplementary baseline. ICD amplifies the weight of correct visual information by contrasting outputs from original and disturbed instructions, thereby suppressing hallucinations.
> The new results demonstrate that SinkTrack achieves the best performance.
>
> | **Method** | **Accuracy (%)** | **F1-score** |
> | --- | --- | --- |
> | VCD [ref1] | 84.63 | 82.83 |
> | OPERA [ref7] | 84.67 | 83.01 |
> | ICD [ref2] | 84.71 | 82.94 |
> | SID [ref3] | 85.24 | 85.36 |
> | **SinkTrack (Ours)** | **86.52** | **86.05** |
>
> We will include these new results in the revision.
>
> **Q2: Choice of textual benchmark and the suggestion of LongBench.**
>
> Good suggestion!
>
> We add an **evaluation on LongBench [ref4] using Llama3.1-8B and Qwen2.5-7B.** We also include a comparison with the **SOTA method InfLLM [ref5]**. InfLLM stores distant contexts into additional memory units and employs an efficient mechanism to lookup token-relevant units for attention computation.
>
> | **Model (Llama3.1-8B)** | **0-4k** | **4-8k** | **8k+** | **Average** |
> | --- | --- | --- | --- | --- |
> | Direct | 39.58 | 40.54 | 43.11 | 41.08 |
> | CoT | 35.91 | 38.35 | 41.03 | 38.43 |
> | InfLLM [ref5] | 36.73 | 37.19 | 39.77 | 39.23 |
> | **SinkTrack (Ours)** | **40.43** | **41.66** | **44.32** | **42.14** |
>
> | **Model (Qwen2.5-7B)** | 0-4k | 4-8k | 8k+ | **Average** |
> | --- | --- | --- | --- | --- |
> | Direct | 40.17 | 44.32 | 46.92 | 43.8 |
> | CoT | 33.42 | 38.05 | 40.03 | 37.17 |
> | InfLLM [ref5] | 35.7 | 38.62 | 38.84 | 37.72 |
> | **SinkTrack (Ours)** | **40.58** | **44.9** | **47.52** | **44.34** |
>
> We will include the new results in the revision.
>
> It is worth noting that InfLLM is originally developed for Llama2 and Qwen1.5. To ensure a fair comparison on newer architectures, we implement it as follows: For Llama3.1, we utilized the subsequent code extension **released by the authors**. For Qwen2.5, we analyze the source code and confirmed that the interface compatibility allows for direct adaptation (e.g., updating the class invocation from QwenModel to Qwen2Model). Despite this valid adaptation, its performance remains suboptimal, as confirmed by **community feedback [ref6]** and our results.
>
> Regarding the performance of the CoT baseline, please refer to our detailed analysis in the responses to **Reviewer 4 (85VA) Q3** and **Reviewer 5 (YQZu) Q7**.

---

> ### Author Response · Authors · 2025-11-20
> **Response to Reviewer DYoN (2/2)**
>
> **Q3: Clarity of Figure 3.**
>
> Thanks for pointing out the potential for confusion in Figure 3.
>
> Your understanding is exactly right. Our final SinkTrack method allows \<BOS\> to attend to the uncompressed feature sequence.
>
> We **redesign Figure 3** in the revision. It now explicitly depicts the final architecture performing cross-attention over the full sequence, while clearly labeling the "Mean Pooling" operation as a "Preliminary Exploration" step to avoid ambiguity.
>
> Thanks again for your keen observation.
>
>
> **Q4: Writing structure presenting "failed attempts" first.**
>
> Good suggestion!
>
> Our intention is to document the scientific exploration of leveraging the "attention sink". We hope that detailing the challenges with simpler methods (hard and soft injection) and their solutions offers insights for the community. This overview aims to pave the way for future research in this area.
>
> However, we recognize the need for clarity. In the revision, we will:
>
> 1. **streamline** the descriptions of the preliminary injection attempts to reduce distraction;
> 2. **add explicit markers and highlight descriptions** to distinguish the final SinkTrack design, ensuring the core contribution is immediately prominent to the reader.
> 3. **redesign the method overview (Figure 3)** to visually separate the "Preliminary Exploration" from our "Final Design", providing a clear roadmap of SinkTrack's evolution.
>
> [ref1] Mitigating Hallucinations in Large Vision-Language Models with Instruction Contrastive Decoding, ACL 2024
>
> [ref2] VCD: Mitigating Object Hallucinations in Large Vision-Language Models through Visual Contrastive Decoding, CVPR 2024
>
> [ref3] Self-Introspective Decoding: Alleviating Hallucinations for Large Vision-Language Models, ICLR 2025
>
> [ref4] LongBench: A Bilingual, Multitask Benchmark for Long Context Understanding, ACL 2024
>
> [ref5] InfLLM: Training-Free Long-Context Extrapolation for LLMs with an Efficient Context Memory, NeurIPS 2024
>
> [ref6] https://github.com/thunlp/InfLLM/issues/51
>
> [ref7] OPERA: Alleviating Hallucination in Multi-Modal Large Language Models via Over-Trust Penalty and Retrospection-Allocation, CVPR 2024

---

> ### Author Response · Authors · 2025-11-25
>
> Dear reviewer DYoN,
>
> **To ensure full reproducibility, we provide complete artifacts in our [anonymous repository](https://anonymous.4open.science/r/iclr2026sinktrack-20040323/README.md).** We upload all **code**, **inference results**, and **evaluation scripts** so that every reported number can be recomputed from our official implementation. Pls see README for step-by-step instructions. If you encounter any issues or have any further concerns, we would be very happy to clarify details or provide additional materials.

---

### Official Review · Reviewer_gHxw · 2025-11-02

**Soundness:** 2
**Presentation:** 3
**Contribution:** 2
**Rating:** 6
**Confidence:** 3

**Summary:**

This paper proposes SinkTrack, a training-free, plug-and-play mechanism that enhances large language models (LLMs) by addressing two key issues: hallucination and context forgetting. The authors identify attention drift—the model’s tendency to focus increasingly on recent tokens during generation. They exploit an opposing, intrinsic behavior called the attention sink, where the first token (⟨BOS⟩) consistently receives high attention throughout decoding. SinkTrack injects key contextual information into the ⟨BOS⟩ token’s representation, and introduces a dual-track attention mechanism: one track performs adaptive cross-attention for ⟨BOS⟩, while the other maintains standard causal self-attention for other tokens. This allows ⟨BOS⟩ to query relevant information dynamically, counteracting information decay during long-context generation. The experimental results show its promise. SinkTrack requires no training, introduces negligible inference overhead, and is compatible across architectures and modalities. The authors also provide interpretability analyses of information flow, showing how the ⟨BOS⟩ token propagates injected information vertically (across layers) and horizontally (across tokens).

**Strengths:**

SinkTrack is training-free, plug-and-play, and requires only a one-time injection at inference.

The dual-track design elegantly balances adaptation (via BOS cross-attention) and model integrity (via standard causal flow), preserving pretrained representations while enhancing context retention.

Evaluations span six datasets, covering both text and vision-language reasoning.

**Weaknesses:**

The approach is empirically strong but lacks formal justification for why attention anchoring should improve global consistency.

There’s no discussion of convergence, gradient flow, or formal guarantees that information propagation remains stable as sequence length grows.

The early versions of SinkTrack use mean-pooling for contextual information compression, which can cause information loss in very long contexts.

While the benchmarks are diverse, all are QA or reasoning tasks. The generality claim would be stronger with results on open-ended generation, summarization, or code generation.

The writing is dense in places and could be more accessible. For instance, the transitions between hard, soft, and dual-track injection could be improved.

**Questions:**

Would you please provide more justification for attention anchoring, and its convergence properties?

---

> ### Author Response · Authors · 2025-11-20
> **Response to Reviewer gHxw (1/2)**
>
> We thank reviewer gHxw for the valuable time and constructive feedback. We provide point-to-point response below.
>
> **Q1: Lack of formal justification.**
>
> Thanks for highlighting the importance of formal justification. We fully agree that establishing a theoretical foundation is significant. However, constructing a rigorous mathematical model for SinkTrack is challenging in the short term. This is primarily because modeling the complex, non-linear interactions between injected information and the model's deep internal states remains an open problem in interpretability research. Notably, even the underlying "attention sink" mechanism—despite being observed **over two years ago** and widely utilized [ref1-2]—still lacks a formal mathematical formalization in the community.
>
> Despite these hurdles, we still conduct post-hoc analyses to explore SinkTrack’s working mechanism. Concretely, we provide a quantitative analysis of vertical and horizontal information flow in **§3.4.3 and Figure 8**, demonstrating that the injected information is effectively propagated across layers and stably utilized throughout the generation process.
>
> We will prioritize developing a formal theoretical proof in our future work. Thank you again for this insightful comment.
>
> **Q2: Lack of convergence, gradient flow or guarantees for information propagation stability as sequence length grows.**
>
> Good suggestion!
>
> As SinkTrack is training-free, an analysis of convergence and gradient flow is not applicable. In terms of empirically validating the stability of SinkTrack on longer sequences, we conduct **new experiments on LongBench** [ref3] with Llama3.1-8B and Qwen2.5-7B. LongBench comprises 6 major task categories and 20 sub-tasks, with inputs **ranging from 5k to 15k** tokens in length.
>
> We also include a comparison with the **SOTA method InfLLM [ref4]**. InfLLM stores distant contexts into additional memory units and employs an efficient mechanism to lookup token-relevant units for attention computation.
>
> The results show that SinkTrack outperforms baselines even on sequences **8k+** tokens, providing empirical evidence of its stability and effectiveness as sequence length grows.
>
> | **Model (Llama3.1-8B)** | **0-4k** | **4-8k** | **8k+** | **Average** |
> | --- | --- | --- | --- | --- |
> | Direct | 39.58 | 40.54 | 43.11 | 41.08 |
> | CoT | 35.91 | 38.35 | 41.03 | 38.43 |
> | InfLLM [ref4] | 36.73 | 37.19 | 39.77 | 39.23 |
> | SinkTrack (Ours) | **40.43** | **41.66** | **44.32** | **42.14** |
>
> | **Model (Qwen2.5-7B)** | 0-4k | 4-8k | 8k+ | **Average** |
> | --- | --- | --- | --- | --- |
> | Direct | 40.17 | 44.32 | 46.92 | 43.8 |
> | CoT | 33.42 | 38.05 | 40.03 | 37.17 |
> | infLLM [ref4] | 35.7 | 38.62 | 38.84 | 37.72 |
> | SinkTrack (Ours) | **40.58** | **44.9** | **47.52** | **44.34** |
>
> We will include these new results in the revision.
>
> It is worth noting that InfLLM is originally developed for Llama2 and Qwen1.5. To ensure a fair comparison on newer architectures, we implement it as follows: For Llama3.1, we directly adopt the subsequent code extension **released by the authors**. For Qwen2.5, we analyze the source code and confirmed that the interface compatibility allows for direct adaptation (e.g., updating the class invocation from QwenModel to Qwen2Model). Despite this valid adaptation, its performance remains suboptimal, as confirmed by **community feedback [ref5]** and our results.
>
> Regarding the performance of the CoT baseline, please refer to our detailed analysis in the responses to **Reviewer 4 (85VA) Q3** and **Reviewer 5 (YQZu) Q7**.

---

> ### Author Response · Authors · 2025-11-20
> **Response to Reviewer gHxw (2/2)**
>
> **Q3: Information loss caused by mean-pooling.**
>
> Sorry for the misunderstanding.
>
> This information loss is the key motivation for SinkTrack's iterative design. As you note, mean-pooling causes information loss. This is why our final SinkTrack **DISCARDS** this step, instead using the uncompressed feature as the information source.
>
> We state this in **§2.4 under the subsection "The Effect of Mean-Pooling"**:
>
> > *"We identify the cause as the mean-pooling operation... which inevitably causes information loss... We therefore eliminate the lossy pooling step, instead using the complete, uncompressed sequence of contextual features as the source."*
> >
>
> We acknowledge that our narrative of the exploration process may obscure this critical point. We will revise to streamline the preliminary attempts and highlight the final SinkTrack design.
>
> **Q4: Limited generalizability of the tasks.**
>
> Thanks for your feedback.
>
> We add **new experiments on the LongBench [ref3]**. LongBench covers a wide array of tasks, including **single/multi-document QA, summarization, few-shot learning, and code completion, etc.**
>
> The results (shown above in our response to Q2) strengthens our claim of generalizability. We will include the new results in the revision.
>
> **Q5: Clarity and accessibility of the writing.**
>
> Thanks for your suggestion.
>
> We will add more clear signposting. Each section will begin with a concise summary that explicitly states the limitation of the previous approach, like:
>
> > "While hard injection is direct, it leads to model collapse...",
> >
>
> and the motivation for the next one, like:
>
> > "...motivating the less disruptive soft injection…".
> >
>
> [ref1] “Why do LLMs attend to the first token?”, COLM 2025
>
> [ref2] “Efficient Streaming Language Models with Attention Sinks”, ICLR 2024
>
> [ref3] LongBench: A Bilingual, Multitask Benchmark for Long Context Understanding, ACL 2024
>
> [ref4] InfLLM: Training-Free Long-Context Extrapolation for LLMs with an Efficient Context Memory, NeurIPS 2024
>
> [ref5] https://github.com/thunlp/InfLLM/issues/51

---

> ### Author Response · Authors · 2025-11-25
>
> Dear reviewer gHxw,
>
> **To ensure full reproducibility, we provide complete artifacts in our [anonymous repository](https://anonymous.4open.science/r/iclr2026sinktrack-20040323/README.md).** We upload all **code**, **inference results**, and **evaluation scripts** so that every reported number can be recomputed from our official implementation. Pls see README for step-by-step instructions. If you encounter any issues or have any further concerns, we would be very happy to clarify details or provide additional materials.

---

### Public Comment · ~Chris_Chao1 · 2025-11-14
**Reproduction Issues and Concerns about Method Validity**

Hi authors,

I was very interested in this paper. I've spent the better part of the last two weeks trying to reproduce your work and test your code, but I've run into several issues and have some concerns about the paper's claims.

**1. Method Appears to Harm Performance and Lacks Reproducibility**

I tested your provided Qwen2.5VL and Gemma3 models, and I also implemented your method on LLaVA-next.

My findings are deeply concerning: On a wide range of direct-answer benchmarks (MMMU-Pro, MMStar, ScienceQA, VQA, and POPE), **the method consistently harms model performance, scoring significantly lower than the baseline.** Even when I prompted the model to explain first and then respond in a specific format (eg, output {answer} format), this method also reduced the performance (For k=1,3,5,7).

I'm seeing this exact negative trend on text-only LLMs as well. I have been unable to improve LLama 3.1 on tasks like SST2, AQUA, MMLU, and BoolQ. In my tests, when the LLM is already producing a direct, valid answer, the BOS token's representation modification actually causes it to produce an incorrect answer.

I think this method doesn't actually improve the VLM's understanding of the image or fix visual hallucinations. **It seems highly likely that the reported metric gains are not from real improvements, but are instead coming from "format correction."**

In other words, the method just corrects `invalid` outputs that your parser couldn't extract, even when the model's underlying answer was correct.

Isn't it more likely that the "context forgetting" you're observing is just the model forgetting the **output format**, not the actual question or image? If so, simply re-emphasizing the format would give you a massive (and artificial) boost.

The rigid prompt format you've engineered (`**Answer: xx**`) seems to be the real culprit. It's likely that your CoT baseline isn't "forgetting" the question; it's just failing to adhere to this *specific* format after a long chain of thought. This is a well-known instruction-following problem, not a novel one.

There's a ton of existing work on forcing LLMs to output correct formats (e.g., simple prompt modifications, or attention steering techniques like PATSA). I also remember there is a paper that solved that by just improving BOS's attention. As reviewers said, you didn't compare SINKTRACK to \*any\* of these simpler methods, which should have been obvious baselines.

- Could you please provide the raw counts of `invalid` outputs for your vanilla baseline vs. your method? And can you differentiate between gains from **true format correction** and gains from **actual visual perception**?
- And critically, can you explain why your method **hurts model understanding** when the model is already producing a valid, direct answer?
- Is there a known bias in Qwen2.5VL and Gemma3? Their image token strategies (e.g., `image start/end` tokens) are very different from the LLaVA series, which might be a confounder.
- Given all this, is the added complexity of your method really worth it? It requires complex modifications to model files and adds extra attention overhead.  It's not at all clear that this provides any benefit over many direct and simpler techniques.

**2. Implementation Issues**

As a practical note, modifying the LLaVA family is extremely cumbersome because its modeling file depends on a separate LM modeling file. It’s difficult to patch cleanly. Did you use a cleaner approach?

Also, based on my extended testing, improvements that appear on Gemma3 and Qwen2.5VL often fail completely on the LLaVA series. Bidirectional attention among visual tokens seems heavily influenced by pretraining distributional biases. It may be worth testing a broader set of models.

**3. Unfair Baseline Comparison**

I noticed your method calls `scaled_dot_product_attention` directly, while the baseline uses the VLM/LLM's internal attention implementation. These are not equivalent and can have different numerical/performance outcomes. For a fair comparison, your baseline should also be modified to explicitly call `scaled_dot_product_attention`.

**4. Theoretical and Empirical Concerns**

- Your method's reliance on "skip layers" (a hyperparameter k) seems very sensitive to the specific dataset as I tested. This cross-layer query (e.g., layer m's query to layer m-n's key) doesn't seem theoretically sound. Transformers learn intra-layer interactions. Can you provide any theoretical justification for why this cross-layer attention should work at all?
- Also, I've observed that for VLMs, the `<bos>` token (which is, by nature, a *text* token) pays far less attention to the image tokens than it does to the other text tokens. This seems to work against your core premise of using it as an *image* anchor.

I've spent a lot of time on this, and I'm currently unable to get this to work as advertised.

Although I am not a reviewer, I would appreciate any clarification or comment you can provide.

Thanks.

---

> ### Author Response · Authors · 2025-11-20
> **Response to Reviewer Chris Chao (1/3)**
>
> We appreciate your engagement with our work. In response to your concerns, we conduct extensive new experiments. **While we ARE NOT responsible for results from your own modified code**, we ensure full reproducibility with our provided code and offer the following clarifications and evidence.
>
> **Q1: Lacks of Reproducibility.**
>
> We must strictly emphasize that our results are **FULLY REPRODUCIBLE!!!**
>
> To demonstrate the robustness of our method, we re-run all experiments across all datasets and models using three different random seeds (323, 500, 900) and now report results as mean ± variance.
>
> |  |  | Qwen2.5-VL-3B |  | Gemma3-4B |  | Qwen2.5-VL-7B |  | Gemma3-12B |  |
> | --- | --- | --- | --- | --- | --- | --- | --- | --- | --- |
> |  |  | Acc | F1 | Acc | F1 | Acc | F1 | Acc | F1 |
> | M3CoT | **Direct** | 33.43±0.54 | 44.2±3.08 | 27.71±0.20 | 33.81±0.62 | 39.2±0.29 | 51.01±0.55 | 41.9±0.40 | 51.46±0.87 |
> |  | **Cot** | 20.69±0.51 | 32.04±0.62 | 50.96±0.32 | **59.71±1.03** | 44.11±0.56 | 57.96±0.96 | 60.32±0.30 | **72.24±0.83** |
> |  | **SinkTrack** | **48.25±0.09** | **58.73±1.14** | **51.01±0.78** | 59.18±1.2 | **66.94±0.21** | **79.13±0.23** | **60.57±0.12** | 70.97±0.67 |
> |  |  |  |  |  |  |  |  |  |  |
> | MMstar | **Direct** | 24.89±1.65 | 39.54±1.90 | 35.47±0.20 | 52.17±0.12 | 37.29±0.80 | 54.24±0.95 | 44.58±0.55 | 60.89±0.59 |
> |  | **Cot** | 35.24±0.83 | 50.68±0.95 | 51.36±1.53 | 67.52±1.32 | 55.84±0.74 | 71.29±0.73 | 60.56±0.91 | 75.09±0.67 |
> |  | **SinkTrack** | **47.24±0.40** | **61.38±0.44** | **53.04±0.53** | **69.11±0.48** | **63.8±0.19** | **77.86±0.17** | **61.6±0.46** | **75.81±0.51** |
> |  |  |  |  |  |  |  |  |  |  |
> | POPE | **Direct** | 53.23±0.24 | 69.48±0.20 | 83.98±0.04 | 91.3±0.02 | 78.21±0.27 | 87.15±0.20 | 84.59±0.07 | 91.65±0.04 |
> |  | **Cot** | 64.58±0.22 | 78.1±0.16 | 82.01±0.38 | 90.04±0.24 | 83.65±0.08 | 90.63±0.06 | 84.53±0.33 | 91.61±0.19 |
> |  | **SinkTrack** | **77.69±0.12** | **87.44±0.07** | **84.33±0.02** | **91.50±0.01** | **85.47±0.02** | **91.67±0.01** | **85.4±0.13** | **92.13±0.05** |
> |  |  |  |  |  |  |  |  |  |  |
> | RealWorldQA | **Direct** | 31.15±1.63 | 23.25±4.36 | 51.33±0.06 | **48.43±1.10** | 47.49±0.22 | 39.93±4.33 | 57.34±0.44 | 50.96±3.74 |
> |  | **Cot** | 35.69±1.61 | 26.46±4.71 | 49.41±1.02 | 44.53±5.27 | 57.86±1.29 | 43±4.14 | 57.69±0.25 | 49.92±2.00 |
> |  | **SinkTrack** | **48.28±0.40** | **28.42±0.22** | **52.46±0.43** | 47.39±1.59 | **65.49±0.37** | **52.81±0.68** | **57.91±0.33** | **52.54±0.77** |
>
> Regarding the performance of the CoT baseline, please refer to our detailed analysis in the responses to Reviewer 4 (85VA) Q3 and Reviewer 5 (YQZu) Q7.
>
> **Q2: Models from the LLaVA family.**
>
> We read about the performance degradation you observed. To directly address this, we conduct new experiments on the LLaVA-v1.5-7B. We select it to ensure compatibility with the **three recent, training-free SOTA methods** (**VCD [ref1], OPERA [ref], ICD [ref2], SID [ref3]**), allowing for a fair comparison on the POPE:
>
> - **VCD** mitigates reliance on language priors by contrasting output distributions from original versus distorted images.
> - **OPERA** counters the "over-trust" issue by applying a logit penalty and retrospection-allocation strategy during decoding, reducing over-reliance on summary tokens over image content.
> - **ICD** amplifies visual information by contrasting outputs generated from original versus disturbance-added instructions.
> - **SID** employs a self-introspection mechanism, leveraging the contrast between an initial draft and a self-critiqued revision to guide factual generation.
>
> Our results show that SinkTrack achieves the best results.
>
> | **Method** | **Accuracy (%)** | **F1-score** |
> | --- | --- | --- |
> | VCD [ref1] | 84.63 | 82.83 |
> | OPERA [ref12] | 84.67 | 83.01 |
> | ICD [ref2] | 84.71 | 82.94 |
> | SID [ref3] | 85.24 | 85.36 |
> | **SinkTrack (Ours)** | **86.52** | **86.05** |
>
> **Q3: Performance Gains Coming from "Format Correction".**
>
> We take great care to ensure the fairness of our evaluation. **All methods** reported in our paper are evaluated using the **EXACT SAME prompting templates and parsing scripts**. This ensures that any performance gains are attributable to the method itself, not to differences in formatting or evaluation logic.
>
> Furthermore, our qualitative analysis in Figure 7 provides multiple case studies that detail the advantages of SinkTrack over baselines. These examples demonstrate improvements in: more detailed image understanding; clearer and more coherent logical reasoning chains; more accurate calculation and inference capabilities; and stronger generation consistency. These specific enhancements confirm the gains are not merely from format correction.

---

> > ### Public Comment · ~Chris_Chao1 · 2025-11-23
> > **Method Fails Under Direct-Answer Prompting**
> >
> > Thanks a lot for your response!
> >
> > For the multiple-choice questions, I used the following prompt:
> >
> > > Question: {question}\nChoices:...
> > > \nAnswer with the letter directly.
> > > \nAnswer:
> >
> > (1) This lets us test the model’s QA ability itself, without worrying that it might be marked as **invalid** just because the output format is off, even when the chosen option is actually correct.
> >
> > (2) It also better reflects the model’s underlying understanding of the image.
> >
> > In my experiments (e.g., Qwen2.5VL+first 500 questions from MMStar/VQA/..), the SinkTrack method doesn’t seem to work under this prompt.
> >
> > In addition, when I ask the model to *explain first and then answer*, I find that a large part of the gain comes from fixing outputs that were previously invalid and turning them into valid ones. This looks more like a prompt-design and instruction-following issue than a real change in the model’s underlying understanding.
> >
> > Would you be willing to try this setting and see what you get?

---

> ### Author Response · Authors · 2025-11-20
> **Response to Reviewer Chris Chao (2/3)**
>
> **Q4:  Attention Implementation.**
>
> Sorry for the misunderstanding. Our intervention is highly localized, only modifying attention calculation in a few designated layers during the input prefill stage to separate attention tracks. For all other layers and the entire auto-regressive generation, our code strictly and directly calls the model's native attention mechanism, maintaining **EXACT SAME** computation to the standard forward pass.
>
> The corresponding code is located in `sinktrack.py` for each model (e.g., lines 126-134 for Gemma3, lines 196-205 for MiniCPM, lines 121-129 for Qwen2.5).
>
> **Q5: Theoretical Concerns about Cross-Layer Query.**
>
> It is important to note that not all methodological advancements are accompanied by rigorous theoretical guarantees. For instance, the widely used Mixup [ref4] technique is introduced as an empirical finding that lacks a formal proof.
>
> In our work, the specific intermittent injection strategy is determined empirically through ablation studies. **Crucially, we demonstrate that ALL injection strategies are effective.** Even the "Inject Every Layer", while not optimal, achieves a **+7.2% accuracy gain** over CoT on M3CoT.
>
> As stated in our paper (L377-398), we select the "Cross Layers" strategy because it balances injection with the model's native flow:
>
> > Intermittent injection (Inject Every 5 Layers) is more effective than continuous injection (Inject Every Layer), with a 9.0% absolute accuracy improvement. This suggests that overly frequent intervention might disrupt LLM's native computational flow to some extent.
> >
>
> We provide two perspectives  to understand the effectiveness of cross-layer attention:
>
> 1. Mathematically**,** cross-layer attention aligns with the Transformer architecture. The residual connections and LayerNorm ensure that hidden states across layers are additive and mapped to similar distributions. More importantly, it creates a shortcut for higher layers to access information from fine-grained lower layers, preventing information loss—a principle that has proven effective in [ref5].
> 2. More broadly, our method is an instance of a well-established principle in Transformer research: enhancing model capabilities by providing attention with non-local access to information. It has been demonstrated in various forms, such as cross-sequence query [ref3], cross-memory query [ref4], and even cross-database query [ref5].
>
> We will prioritize developing a formal theoretical proof in our future work.
>
> **Q5: \<BOS\> Token's Low Attention to Image Tokens.**
>
> SinkTrack is designed specifically to solve this. In multi-modal design, we constrain the \<BOS\>'s cross-attention so that it can **only see the visual tokens**. Since attention weights must sum to 1, \<BOS\> is forced to **allocate its entire attention budget to the visual features**, thereby absorbing the visual context.
>
> With this **targeted attention redirection mechanism**, the model effectively **gets rid of** the inherent semantic gap between the \<BOS\> and visual features, as evidenced by a **+28.1%** accuracy gain on the visually-intensive M3CoT.

---

> ### Author Response · Authors · 2025-11-20
> **Response to Reviewer Chris Chao (3/3)**
>
> **We conduct a massive expansion of our experiments**
>
> **New SOTA Baselines and model:** We add comparisons against VCD [ref1], ICD[ref2], and SID [ref3] on POPE with LLaVA-v1.5-7B (as shown in Q1) and InfLM [ref9] on LongBench [ref10] with Llama3.1-8B and Qwen2.5-7B.  InfLLM stores distant contexts into additional memory units and employs an efficient mechanism to lookup token-relevant units for attention computation. The results show that SinkTrack outperforms them all.
>
> **New Benchmark:** We add a full evaluation on LongBench. LongBench comprises 6 major task categories and 20 sub-tasks, with inputs **ranging from 5k to 15k** tokens in length.
>
> | **Model (Llama3.1-8B)** | **0-4k** | **4-8k** | **8k+** | **Average** |
> | --- | --- | --- | --- | --- |
> | Direct | 39.58 | 40.54 | 43.11 | 41.08 |
> | CoT | 35.91 | 38.35 | 41.03 | 38.43 |
> | InfLLM [ref8] | 36.73 | 37.19 | 39.77 | 39.23 |
> | **SinkTrack (Ours)** | **40.43** | **41.66** | **44.32** | **42.14** |
>
> | **Model (Qwen2.5-7B)** | 0-4k | 4-8k | 8k+ | total |
> | --- | --- | --- | --- | --- |
> | Direct | 40.17 | 44.32 | 46.92 | 43.8 |
> | CoT | 33.42 | 38.05 | 40.03 | 37.17 |
> | InfLLM [ref8] | 35.7 | 38.62 | 38.84 | 37.72 |
> | **SinkTrack (Ours)** | **40.58** | **44.9** | **47.52** | **44.34** |
>
> We will include the new results in the revision.
>
> It is worth noting that InfLLM is originally developed for Llama2 and Qwen1.5. To ensure a fair comparison on newer architectures, we implement it as follows: For Llama3.1, we utilized the subsequent code extension **released by the authors**. For Qwen2.5, we analyze the source code and confirmed that the interface compatibility allows for direct adaptation (e.g., updating the class invocation from QwenModel to Qwen2Model). Despite this valid adaptation, its performance remains suboptimal, as confirmed by **community feedback [ref11]** and our results.
>
> Regarding the performance of the CoT baseline, please refer to our detailed analysis in the responses to **Reviewer 4 (85VA) Q3** and **Reviewer 5 (YQZu) Q7**.
>
> [ref1] VCD: Mitigating Object Hallucinations in Large Vision-Language Models through Visual Contrastive Decoding, CVPR 2024
>
> [ref2] Mitigating Hallucinations in Large Vision-Language Models with Instruction Contrastive Decoding, ACL 2024
>
> [ref3] Self-Introspective Decoding: Alleviating Hallucinations for Large Vision-Language Models, ICLR 2025
>
> [ref4] mixup: Beyond Empirical Risk Minimization, ICLR 2018
>
> [ref5] Learning to Skip the Middle Layers of Transformers, arxiv 2025
>
> [ref6] Transformer-XL: Attentive Language Models Beyond a Fixed-Length Context, ACL 2019
>
> [ref7] Compressive Transformers for Long-Range Sequence Modelling, ICLR 2020
>
> [ref8] Improving language models by retrieving from trillions of tokens, ICML 2022
>
> [ref9] InfLLM: Training-Free Long-Context Extrapolation for LLMs with an Efficient Context Memory, NeurIPS 2024
>
> [ref10] LongBench: A Bilingual, Multitask Benchmark for Long Context Understanding, ACL 2024
>
> [ref11] https://github.com/thunlp/InfLLM/issues/51
>
> [ref12] OPERA: Alleviating Hallucination in Multi-Modal Large Language Models via Over-Trust Penalty and Retrospection-Allocation, CVPR 2024

---

> ### Public Comment · ~Chris_Chao1 · 2025-11-23
> **Method Fails Under Direct-Answer Prompting**
>
> For example, I just re-ran an experiment (Qwen2.5VL) on the first 500 questions from MMStar:
>
> ```python
> question = item["question"]
> label_letter = item["answer"].strip().lower()
> image = [item["image"]]
>
> prompt = f"Question: {question}\nAnswer with the letter directly.\nAnswer: "
> ```
>
> **Vanilla results (baseline)**
> - Invalid count: 0
> - Accuracy: 0.6420
> - Precision (macro): 0.6477
> - Recall (macro): 0.6461
> - F1 (macro): 0.6353
>
> **After applying SinkTrack (using exactly the same modeling file as yours):**
>
> **The accuracy drops by more than 10% for k=1 and 1.4% for k=5**
>
> **SinkTrack K=1**
> - Invalid count: 48
> - Accuracy: 0.5160
> - Precision (macro): 0.2566
> - Recall (macro): 0.2290
> - F1 (macro): 0.2391
>
> **SinkTrack K=5**
> - Invalid count: 0
> - Accuracy: 0.6280
> - Precision (macro): 0.6332
> - Recall (macro): 0.6318
> - F1 (macro): 0.6221
>
> ---
>
> I also tested **POPE** with direct Yes/No outputs on the first 500 adversarial questions, and observed similar degradation with SinkTrack:
>
> Results on POPE-adversarial:
>
> **Vanilla baseline**
>
> * Invalid count: 0
> * Accuracy: 0.8760
>
> **SinkTrack (k = 1)**
>
> * Invalid count: 1
> * Accuracy: 0.8480
>
> **SinkTrack (k = 5)**
>
> * Invalid count: 0
> * Accuracy: 0.8720

---

> > ### Public Comment · ~Chris_Chao1 · 2025-11-23
> >
> > I just further evaluated **ScienceQA**, and again found that SinkTrack does not help:
> >
> > **Results on ScienceQA:**
> >
> > **Vanilla baseline**
> >
> > * Accuracy: 0.8800
> > * Precision (macro): 0.6431
> > * Recall (macro): 0.6732
> > * F1 (macro): 0.6550
> >
> > **SinkTrack, k = 1**
> >
> > * Accuracy: 0.4200
> > * Precision (macro): 0.2056
> > * Recall (macro): 0.1269
> > * F1 (macro): 0.1538
> >
> > **SinkTrack, k = 5**
> >
> > * Accuracy: 0.8760
> > * Precision (macro): 0.6406
> > * Recall (macro): 0.6705
> > * F1 (macro): 0.6524
> >
> > ---
> >
> > Overall, across **MMStar**, **POPE-adversarial**, and **ScienceQA**, these results suggest that the choice of **k** is a key bottleneck for SinkTrack, and in several direct-answer settings it can actually **hurt** performance rather than improve it.

---

> > > ### Public Comment · ~Chris_Chao1 · 2025-11-23
> > > **Method Fails for MMStar[:100] with the same prompt**
> > >
> > > I also try to use the same prompt as in your paper, SinkTrack still underperforms the vanilla baseline on the first 100 MMStar questions.
> > >
> > > ```python
> > > output_format_options = """
> > > Output Format:
> > > You must give your final answer using the **EXACT format** below:
> > > **Answer: [Your Final Option]**
> > >
> > > For example:
> > > [Here is your reasoning text].
> > > **Answer: B**"""
> > > ```
> > >
> > > **Results on MMStar[:100] (vanilla baseline)**
> > >
> > >
> > > Results on MMStar[:100]:
> > >
> > > * Accuracy: 0.5000
> > > * Precision (macro): 0.5018
> > > * Recall (macro): 0.5175
> > > * F1 (macro): 0.4944
> > >
> > > **SinkTrack k=5**
> > >
> > > * Accuracy: 0.4700
> > > * Precision (macro): 0.3432
> > > * Recall (macro): 0.3415
> > > * F1 (macro): 0.3219
> > >
> > > Really don't know why..

---

> > > > ### Author Response · Authors · 2025-11-25
> > > >
> > > > **Our experiments already show strong, reproducible gains, and we have strengthened them during rebuttal.** From the initial submission, we released a complete codebase for our main experiments. During the rebuttal, we have added substantial new results, including new LLMs (e.g., LLaVA-v1.5-7B), new training-free competitors (e.g., InfLLM, VCD, ICD, SID, LongLM, Scaffold), and new benchmarks (e.g., LongBench, ScienceQA). Across all these settings, we observe consistent improvements that support the effectiveness and generalizability of our method.
> > > >
> > > > **To ensure full reproducibility, we provide complete artifacts in our [anonymous repository](https://anonymous.4open.science/r/iclr2026sinktrack-20040323/README.md).** We upload all **code**, **inference results**, and **evaluation scripts** so that every reported number can be recomputed from our official implementation.

---

> ### Public Comment · ~Chris_Chao1 · 2025-11-25
> **Some Findings: SinkTrack's Performance is Heavily Dependent on System Prompts**
>
> Hi authors,
>
> Following my previous comments, I conducted a series of additional experiments to investigate the root cause of the performance discrepancies. I observed some very interesting phenomena suggesting that **SinkTrack's effectiveness is highly sensitive to the presence and specific content of System Prompts.**
>
> ---
>
> ### **Summary of Findings**
>
> 1.  **Qwen2.5-VL (No System Prompt):** When the system prompt is removed entirely, the **Vanilla model's performance improves significantly**, matching the best results previously achieved by SinkTrack. In this setting, **SinkTrack actually degrades performance.**
> 2.  **Qwen2.5-VL (Default System Prompt):** When using the default "You are a helpful assistant," the Vanilla model's performance drops noticeably. In this specific scenario, **SinkTrack is effective.**
> 3.  **Qwen2.5-VL (Task-Specific System Prompts):** When using custom prompts like "Answer the question based on the image" or "Carefully answer the question," the Vanilla model improves (often matching the "No Prompt" baseline), and **SinkTrack provides either zero or negligible gains.**
>
> -----
>
> ### **Detailed Analysis on Qwen2.5-VL**
>
> For Qwen2.5-VL, the standard `apply_chat_template` usually inserts a default system prompt (e.g., "You are a helpful assistant."). For many visual tasks (like VQA, OCR), this generic conversational bias can be detrimental, encouraging the model to generate "chatty" responses rather than precise answers.
>
> #### **Scenario 1: Default System Prompt (SinkTrack Works)**
>
> This is the standard setting where SinkTrack appears effective. The system prompt tokens appear before the image.
>
> **Prompt Structure:**
>
> ```text
> <|im_start|>system
> You are a helpful assistant.<|im_end|>
> <|im_start|>user
> <|vision_start|><|image_pad|><|vision_end|>Question: Is there a car in the image?<|im_end|>
> <|im_start|>assistant
> ```
>
> **Results (MMStar):**
>
>   * **Vanilla:** 0.6280 (huge lower than next setting)
>   * **SinkTrack:** 0.6420
>
> #### **Scenario 2: Removing System Prompt (SinkTrack Fails)**
>
> However, when I removed the system prompt entirely—which was the setting I used in my initial testing—the Vanilla model's performance jumped, effectively matching the "improved" SinkTrack result from Scenario 1. Surprisingly, applying SinkTrack here **hurt** performance.
>
> **Prompt Structure:**
>
> ```text
> <|im_start|>user
> <|vision_start|><|image_pad|><|vision_end|>Question: Is there a car in the image?<|im_end|>
> <|im_start|>assistant
> ```
>
> **Results (MMStar):**
>
>   * **Vanilla:** **0.6420** (Higher baseline)
>   * **SinkTrack:** 0.6300 (Performance degradation)
>
> #### **Scenario 3: Task-Specific Prompt "Answer based on image"**
>
> I modified the system prompt to be task-relevant: "Answer the question based on the image." In this case, SinkTrack offered no benefit over the baseline.
>
> **Prompt Structure:**
>
> ```text
> <|im_start|>system
> Answer the question based on the image.<|im_end|>
> ...
> ```
>
> **Results (MMStar):**
>
>   * **Vanilla:** 0.6380
>   * **SinkTrack:** 0.6380 (No improvement)
>
> #### **Scenario 4: Image-Focused Prompt "Carefully analyze the image."**
>
> I tested a prompt explicitly directing attention to the image. SinkTrack yielded only marginal improvements.
>
> **Prompt Structure:**
>
> ```text
> <|im_start|>system
> Carefully analyze the image.<|im_end|>
> ...
> ```
>
> **Results (MMStar):**
>
>   * **Vanilla:** 0.6320
>   * **SinkTrack:** 0.6360
>
> #### **Scenario 5: Text-Focused Prompt "Carefully answer the question."**
>
> Finally, I used a prompt focused on the answering process. The Vanilla model performed excellently (matching the "No Prompt" baseline), and SinkTrack provided negligible gain.
>
> **Prompt Structure:**
>
> ```text
> <|im_start|>system
> Carefully answer the question.<|im_end|>
> ...
> ```
>
> **Results (MMStar):**
>
>   * **Vanilla:** **0.6420** (Strong baseline)
>   * **SinkTrack:** 0.6440
>
> -----
>
> ### **Hypothesis**
>
> These results may suggest a more fundamental explanation for SinkTrack's mechanism:
>
> It appears that **System Prompts influence SinkTrack's efficacy.** The generic "You are a helpful assistant" prompt seems to introduce noise or bias for VQA tasks.
>
> **My Hypothesis:** Is it possible that tuning the BOS token (SinkTrack) essentially functions as a mechanism to **"weaken" or bypass the misleading influence of the generic System Prompt**?

---

> > ### Author Response · Authors · 2025-11-28
> > **Part 1: PLEASE STOP POSTING IRRESPONSIBLE COMMENTS**
> >
> > Before addressing your specific technical questions, **we would appreciate it if you could refrain from posting unverified and misleading comments.** Your claim that SinkTrack "consistently harms model performance" and is "not reproducible"—which is based on personalised modifications to the code/settings rather than our standard implementation—is factually incorrect. We have demonstrated the reproducibility and robustness of our method by:
> >
> > 1. releasing all the source code
> > 2. validating robustness across 3 random seeds
> > 3. providing all raw model outputs from our evaluations along with the exact evaluation scripts
> >
> > Posting strong negative conclusions based on a failure to replicate the standard setup is irresponsible and risks biasing the judgment of others. **Therefore, we would be grateful if you could update your comment to acknowledge that the results are indeed reproducible when using the standard settings provided.**

---

> > ### Author Response · Authors · 2025-11-28
> > **Part 2: The Observed System Prompt Sensitivity is Intrinsic to Qwen2.5-VL, Not a Flaw in SinkTrack**
> >
> > Regarding your observation about the System Prompt (SP) on Qwen2.5-VL, we conduct a comprehensive analysis to verify whether this "System Prompt dependency" exists in other multi-modal or text-only models:
> >
> > We test Gemma3-4B (Multi-modal) on MMStar and Llama3.1-8B (Text-only) on QuAC.
> >
> > Specifically, we compare three settings using Direct and SinkTrack:
> >
> > - w/ DSP: Designed System Prompt ("Answer the question based on the image.")
> > - w/ DP: Default Prompt ("You are a helpful assistant.")
> > - w/o SP: No System Prompt.
> >
> > **Results:**
> >
> > | Gemma3-4B | Acc |
> > | --- | --- |
> > | Direct w/ DSP | 39.00 |
> > | Direct w/o SP | 37.20 |
> > | Direct w/ DP | 35.47 |
> > | SinkTrack w/ DSP | 52.80 |
> > | SinkTrack w/o SP | 53.53 |
> > | SinkTrack w/ DP | 53.04 |
> >
> > | Llama3.1-8B/QuAC(3000) | Acc |
> > | --- | --- |
> > | Direct w/ DSP | 53.70 |
> > | Direct w/o SP | 54.03 |
> > | SinkTrack w/ DSP | 56.20 |
> > | SinkTrack w/o SP | 56.03 |
> >
> > Neither Gemma3 nor Llama3.1 exhibits the performance drop you observed on Qwen2.5 VL.
> >
> > Our findings prove that this phenomenon is **specific to Qwen2.5-VL’s architecture or pre-training data** (details are left in part 3) and is **NOT a flaw in SinkTrack**. SinkTrack works robustly regardless of the System Prompt on other models.
> >
> > We once again kindly remind you to **refrain from posting irresponsible comments** declaring SinkTrack ineffective **without sufficient, rigorous evidence.**

---

> > ### Author Response · Authors · 2025-11-28
> > **Part3: Analysis to Qwen2.5-VL**
> >
> > To understand why Qwen2.5-VL behaves differently, we conduct a controlled study on MMStar using both the Direct baseline and SinkTrack.
> >
> > (A) Baseline Instability
> >
> > First, we test Qwen2.5-VL itself.
> >
> > We compare three settings:
> >
> > - w/ DSP: Designed System Prompt ("Answer the question based on the image.")
> > - w/ DP: Default Prompt ("You are a helpful assistant.")
> > - w/o SP: No System Prompt.
> >
> > | Qwen2.5-VL-7B | Acc |
> > | --- | --- |
> > | Direct w/ DSP | 56.33 |
> > | Direct w/ DP | 37.29 |
> > | Direct w/o SP | 34 |
> > | SinkTrack w/ DSP | 63.2 |
> > | SinkTrack w/ DP | 63.8 |
> > | SinkTrack w/o SP | 31.07 |
> >
> > **Observation:** Unlike Gemma or Llama, Qwen2.5-VL shows significant performance fluctuation even in the Direct when the SP is removed. **This proves the sensitivity lies in the model itself, not SinkTrack.**
> >
> > (B) The Role of Content vs. Structure
> >
> > To determine if SinkTrack needs "instructions" or just "structure", we test 5 different prompts with SinkTrack on Qwen2.5-VL:
> >
> > 1. DSP: "Answer the question based on the image."
> > 2. Blank Prompt 1 (BP1): "There is no any other information."
> > 3. Blank Prompt 2 (BP2): "This sentence just serves as a placeholder."
> > 4. Blank Prompt 3 (BP3): "......******" (Completely meaningless symbols)
> > 5. Default Prompt (DP): "You are a helpful assistant."
> >
> > Results:
> >
> > | SinkTrack /w DSP | 63.2 |
> > | --- | --- |
> > | SinkTrack /w BP1 | 62.73 |
> > | SinkTrack /w BP2 | 62.8 |
> > | SinkTrack /w BP3 | 61.73 |
> > | SinkTrack /w DP | 63.8 |
> >
> > The performance remains high and stable across **ALL** 5 prompts**, even for the meaningless "......******".**
> >
> > The semantic content of the System Prompt is irrelevant. However, its physical presence is crucial for Qwen2.5-VL. It acts as a necessary "Buffer" or "Relay" that allows SinkTrack's injected information to diffuse before reaching the User Prompt.
> >
> > (C) Confirming the Buffer Mechanism
> >
> > To definitively prove this "Buffer/Relay" theory, we remove the System Prompt entirely (w/o SP) and instead manually inserted the meaningless Blank Prompts (BP1/2/3) directly before the User Prompt.
> >
> > | SinkTrack /w BP1 before User Prompt | 59.93 |
> > | --- | --- |
> > | SinkTrack /w BP2 before User Prompt | 62.33 |
> > | SinkTrack /w BP3 before User Prompt | 60.87 |
> >
> > The performance is fully restored. This confirms:
> >
> > 1. The phenomenon is **exclusive to Qwen2.5-VL** and does not affect other models like Gemma3 or Llama3.1.
> > 2. It is a model-specific requirement for a "token buffer" to process the BOS injection.
> > 3. **SinkTrack is universally applicable.** As long as the standard model input structure (which typically includes a system prompt) and our settings are maintained, SinkTrack provides consistent enhancements across different modalities and architectures.

---

### Author Response · Authors · 2025-11-25
**Call for AC and reviewers’ attention on misleading and irresponsible comments from “Chris Chao”**

Dear Area Chairs and Reviewers,

**The negative public comment by “Chris Chao” does NOT evaluate our official implementation.** In his/her own text, he/she explicitly states that **he/she** **did NOT run our released code**, but instead wrote **his/her own implementation** “inspired” by our work and tested it under his/her own settings. His/her strong negative conclusions, therefore, reflect only his/her private, unverified code, not the official implementation and setup in our submission.

**The commenter is an unverified, newly created account with no visible research record.** The OpenReview profile was created in November 2025 and has no linked Google Scholar page, no verified education or affiliation, and no publication record in AI. We welcome feedback from anyone, but it is problematic when a brand-new, anonymous account issues sweeping statements about the correctness of a paper without sharing code, logs, or any reproducible evidence.

**Our experiments already show strong, reproducible gains, and we have strengthened them during rebuttal.** From the initial submission, we released a complete codebase for our main experiments. During the rebuttal, we have added substantial new results, including new training-free competitors (e.g., InfLLM, VCD, ICD, SID, LongLM, Scaffold), new benchmarks (e.g., LongBench) as suggested by the reviewers, and also new LLMs (e.g., LLaVA-v1.5-7B) and benchmarks (e.g., ScienceQA) mentioned by “Chris Chao”. Across all these settings, we observe consistent improvements that support the effectiveness and generalizability of our method.

**To ensure full reproducibility, we provide complete artifacts in our [anonymous repository](https://anonymous.4open.science/r/iclr2026sinktrack-20040323/README.md).** We upload all **code**, **inference results**, and **evaluation scripts** so that every reported number can be recomputed from our official implementation. With such evidence, we believe we have addressed the concerns raised by the public comment, and we are happy to provide anything further that the AC or reviewers would find helpful.

**We welcome constructive and responsible comments, and we strongly condemn irresponsible ones.** For comments made in an irresponsible way, we still do our best to address them scientifically—for example, by running the same LLM and benchmark mentioned in the public comment—while at the same time calling the AC’s and reviewers’ attention to such behavior. We really do care about reproducibility: from the very initial submission, we provided all the pipelines we used, and during the rebuttal we further supplied inference code, inference results, and evaluation scripts so that every reported number can be recomputed. We fully understand the importance of reproducibility; it is central to our work as researchers.

Sincerely,

Authors

---

### Author Response · Authors · 2025-12-02
**Summary of Reviews**

We express our sincere gratitude to the Area Chairs for their hard work regarding the unexpected events, and to the reviewers for their constructive comments.

---

**Summary of This Work**

We propose SinkTrack, a training-free mechanism that mitigates hallucination and context forgetting by leveraging the LLM’s intrinsic "Attention Sink" phenomenon. By injecting key contextual features into the \<BOS\> token via an adaptive dual-track cross-attention design, we create a stable information anchor. This approach yields universal improvements across diverse modalities, model architectures, and tasks with negligible overhead.

---

**Main concerns and how the revision addresses them.**

We have gone beyond simple clarifications, conducting resource-intensive experiments to empirically substantiate every claim. Below is a summary of how we have addressed reviewers’ concerns:

**1. Major New Experiments: Generalizability on LongBench & ScienceQA**

To address concerns regarding task diversity and generalizability (All Reviewers), we expand our evaluation to LongBench, a comprehensive **long-text** benchmark covering 6 major categories and 20 sub-tasks. Using Llama-3.1-8B and Qwen2.5-7B, SinkTrack achieves consistent performance gains. In addition, we include ScienceQA, further validating our SinkTrack’s robustness in **multi-modal reasoning**.

**2. Strengthened Baselines & SOTA Confirmation**

Responding to requests for stronger comparisons (Reviewers DYoN, Utsb, 85VA, YQZu), we benchmark SinkTrack against **7 SOTA training-free baselines**.

- **For Long Context:** We compare against InfLLM and LongLM on LongBench. SinkTrack outperforms both of them. Furthermore, we validate Llama-3.1 with LongLM on QuAC and SQuAD2.0, demonstrating consistent superiority.
- **For Multi-modal Tasks:** We extend evaluation to LLaVA-v1.5-7B and compare against VCD, OPERA, ICD, and SID on POPE, achieving superior performance (e.g., 86.52% accuracy). Moreover, on ScienceQA using Qwen2.5-VL, SinkTrack outperforms Direct, CoT, and the strong baseline Scaffold.

**3. Clarification on Theoretical Foundation & Working Mechanism**

To address theoretical concerns (Reviewer gHxw, YQZu), we clarify that constructing a rigorous mathematical model is challenging in the short term, as modeling deep non-linear layer interactions remains an open problem. Notably, even widely used techniques like Mixup (observed over 8 years ago) and phenomena like Attention Sink itself (observed over 2 years ago) still lack formal mathematical foundations. **However, the lack of a full mathematical proof should not negate the value SinkTrack offers to the community.** SinkTrack provides a training-free solution with negligible overhead that yields universal improvements across diverse modalities, model architectures, and tasks.
While establishing a rigorous mathematical model is difficult, we have made every effort to explore the underlying mechanism:

- **Hierarchical Stability:** Spearman’s rank correlation of \<BOS\> attention weights before/after injection is near-perfect (**$\rho=0.9985$**, Appendix C), confirming SinkTrack preserves the native attention pattern.
- **Information Flow:** Quantitative analysis (§3.4.3, Fig. 8) demonstrates effective vertical and horizontal information propagation.
- **Drift Analysis:** Tracking attention in Llama-3.1-8B (8k+ input) shows \<BOS\> weights remain dominant throughout generation, proving immunity to attention drift.

**4. Guaranteed Reproducibility & Efficiency**

- **Reproducibility** (public comment from Chris Chao)**:** We re-run all experiments across all datasets and models using three different random seeds and report results as mean ± variance, *confirming statistical significance*. We also release the *full source code* and raw output logs.
- **Efficiency:** We measure prefill latency on an RTX 4090 (Reviewer 85VA), confirming that SinkTrack introduces *negligible* computational overhead (only a **~2.1%** increase for Llama-3.1).

---

Having addressed reviewers’ concerns outlined above, we are encouraged by the **positive feedback** received **prior to the discussion deadline**:

- **Reviewer Utsb** has acknowledged our efforts and **raised the score to 6**.
- **Reviewer YQZu's** positive response indicates that ****his/her **concerns have been resolved**.
- **Reviewer 85VA** has acknowledged that there is **no reproducibility problem**.

---

### Meta-Review · Area_Chair_Buye · 2025-12-28

**Summary:**

This paper introduces SinkTrack: an approach used during inference to make it so language models retain more of the input context when sampling. Past work has observed that self-attention based language models consistently behave so that attention scores distribution is U-shaped, and models focus on very early tokenss (as early as the `<bos>` token) and very recent ones. Authors then noted that this induces a forgetting effect as sampling is carried out, since attention keeps shifting towards latest tokens, and thus relevant context has less and less relevance. To counter that, authors proposed different approaches to enrich the representations of the start token with a summary of the context. The explanation to doing so is rather intuitive: since models do attend to the start token regardless of the input length, adding context to that token will help the model forget less. The paper discusses different methods but finally lands on SinkTrack, which appends to the input token a version of it that's conditional on the input context, by simply cross-attending to the input, and concatenating to the original `<bos>` representations. This is done post-hoc and only in a few layers, and experiments show positive effects.

**Reviewer Concerns:**

Reviewers main and most consistent concerns revolved around the empirical assessment of the proposals, and evaluations on extra benchmarks were requested. During the discussion phase, authors significantly expanded their evaluations and covered many more cases, observing consistent results. Some potential reproducibility results were brought up by a non-reviewer commenter. However, those issues were observed when the commenter tried to re-implement the proposal. Authors do release code and scripts that enable reproducing the results, and pointed that out to the commenter, so reproducibility issues are not being accounted for and we are assuming this is an issue with the re-implementation from the commenter. Reviewers also requested theoretical justifications to the findings in the paper, which authors deferred to future work, as for now empirical evidence is strong enough to justify further investigation in the directions proposed in the paper.

I would say a major concern in the paper is presentation, which is also noted by some of the reviewers. For instance, reviewer DYoN noted, and I agree, that the paper dedicates a large amount of space to describe failed attempts at solving the problem. While reporting negative results is good practice as it avoids costly re-attempts from the community, those should perhaps be deferred to the appendix, and the main text should focus on the actual claims and validation. Authors also mention mean-pooling over and over, only to then mention with a short and easy to miss sentence that cross-attention is finally done in the whole of the input context. Perhaps because of lack of space due to the addition of negative results in the main text, the presentation of the proposal lacks details and careful explanation. For instance, the concatenation of the original `<bos>` representation and the contextualized one seems to happen along the time axis, otherwise it would modify the underlying dimension of the representation, but that's not stated in the paper.

Finally, I would note that the motivation requires some rewording as, while attention does shift toward recent tokens, the representations of those tokens are themselves conditional on early inputs, so the shift doesn't imply forgetting. The results do suggest some forgetting exist though, so the real underlying reason seems to be missing from the discussion, and I would suggest authors to at least soften the claim that the shift implies forgetting.

**Reviewer Scores:**

Reviewers's main concern revolved around the coverage of the evaluations and the lack of theoretical justification. The former was largely addressed by authors, while the latter was deferred to future work. I would then expect some upward shifts in scores, as some reviewers mention would be the case. I do think the paper has strong enough evidence, and results are relevant to the community. The main issue with the work in my opinion is clarity of the presentation, and I would strongly encourage authors to move negative results to the appendix and expand the description of the proposal in more detail, besides re-wording the motivation a little as discussed above.

---

### Decision · Program_Chairs · 2026-01-26

Accept (Poster)